# Blood circulation of soft nanomaterials is governed by dynamic remodeling of protein opsonins at nano-biointerface

Srinivas Abbina [1,2,7], Lily E. Takeuchi[1,2,7], Parambath Anilkumar [1,2,7], Kai Yu[1,2], Jason C. Rogalski[3], Rajesh A. Shenoi[4], Iren Constantinescu[1,2] & Jayachandran N. Kizhakkedathu [1,2,5,6 ✉]

Nanomaterials in the blood must mitigate the immune response to have a prolonged vascular residency in vivo. The composition of the protein corona that forms at the nano-biointerface may be directing this, however, the possible correlation of corona composition with blood residency is currently unknown. Here, we report a panel of new soft single molecule polymer nanomaterials (SMPNs) with varying circulation times in mice ($t_{1/2\beta}$ ~ 22 to 65 h) and use proteomics to probe protein corona at the nano-biointerface to elucidate the mechanism of blood residency of nanomaterials. The composition of the protein opsonins on SMPNs is qualitatively and quantitatively dynamic with time in circulation. SMPNs that circulate longer are able to clear some of the initial surface-bound common opsonins, including immunoglobulins, complement, and coagulation proteins. This continuous remodelling of protein opsonins may be an important decisive step in directing elimination or residence of soft nanomaterials in vivo.

[1] Centre for Blood Research, Life Sciences Institute, The University of British Columbia, Vancouver, BC V6T 1Z3, Canada. [2] Department of Pathology and Laboratory Medicine, The University of British Columbia, Vancouver, BC V6T 2B5, Canada. [3] Centre for High Throughput Biology, Michael Smith Laboratories, The University of British Columbia, Vancouver, BC V6T 1Z1, Canada. [4] Inter University Centre for Biomedical Research & Super Speciality Hospital, Mahatma Gandhi University Campus at Thalappady, Rubber Board P O, Kottayam, Kerala 686 009, India. [5] Department of Chemistry, The University of British Columbia, Vancouver, BC V6T 1Z1, Canada. [6] School of Biomedical Engineering, The University of British Columbia, Vancouver, BC V6T 1Z3, Canada. [7] These authors contributed equally: Srinivas Abbina, Lily E. Takeuchi, Parambath Anilkumar. ✉email: jay@pathology.ubc.ca

Nanomaterials are cleared from the blood by the mononuclear phagocyte system[1,2]. The interaction of nanomaterials with circulating blood components is crucial in governing their biological fate and functions and is highly relevant to biocompatibility and toxicity[2]. Studies using a variety of nanomaterials, including liposomes, micelles, inorganic/organic hard nanoparticles, polymers, and 'self' peptide conjugated nanoparticles have been performed to predict how physiochemical characteristics influence the blood residency or immune recognition of nanomaterials[2–5]. However, there is a still lack of fundamental understanding on how some nanomaterials are eliminated rapidly from the blood and accumulate in organs, while others achieve long residency in blood[6]. A detailed understanding of this fundamental phenomenon is highly useful in generating long acting therapeutics and farther our understanding on the biocompatibility/toxicity of nanomaterials.

Rapid deposition of protein opsonins on nanomaterial's surface upon introduction into the blood is well established and believed that they control the immune recognition of nanomaterials[7–13]. Some key insights into the differences in protein composition at the interface with time, size, and surface chemistry of nanomaterials have been reported in vitro to understand the immune evasion of nanomaterials, however sparse information is available on this nano-biointerface inside the body[11,14–16]. Moreover, it is vital to investigate the unanswered questions including whether the nano-biointerface is dynamic in vivo or the initially adsorbed protein opsonins are the de facto characteristics of a system that guides its fate in circulation. Importantly, biological systems are highly responsive to external stimuli, such as the introduction of nanomaterials, which could continuously alter the material–protein interactions and may be functionally relevant to protein opsonin changes. Thus, a thorough investigation of the evolution of proteins at the nano-biointerface in vivo over extended time periods could provide clues into this unresolved puzzle. Many of the currently used systems, both in vitro and in vivo, do not qualify as long circulating nanomaterials to assess their in vivo protein corona due to their poor chemical and biological stability[11,15,17]. Highly biocompatible and biologically stable nanomaterials, can be easily separated from blood, are very much desirable to perform such studies, which have not been previously explored.

In this work, we developed a class of highly hydrophilic, biocompatible, soft single molecule polymer nanomaterials (SMPNs) with different blood circulation profiles (short to ultra-long) while maintaining similar surface chemistry to uncover the interplay between the nature of nano-biointerface and blood residency in vivo over clinically relevant time scales. We performed unbiased tandem mass spectrometry-based proteomics to reveal the evolution of protein composition with time on SMPNs in vivo and their fate in circulation.

## Results

**Blood circulation of SMPNs.** A multitude of nanomaterial characteristics including surface chemistry, hydrophilicity, size and shape, charge, rigidity, stability, and biocompatibility are detrimental to achieve long blood circulation or its susceptibility to accumulate in organs[18–20]. In particular, soft and non-fouling materials showed the promise of avoiding opsonization. We developed three mega hyperbranched polyglycerol based SMPNs by the polymerization of glycidol (Supplementary Table 1)—SMPN-1, SMPN-3, and SMPN-9—named after their molecular weight ($M_w$), 1.3, 2.9, and 9.3 million Daltons, respectively (Fig. 1a, b, Supplementary Fig. 1)[21]. The SMPNs are very compact, hydrophilic, stable, and biocompatible. They have neutral surface charge and possess a high number of functionalizable end

groups (Fig. 1a, b and Supplementary Fig. 2). The average hydrodynamic sizes of the SMPN-1, −3, and −9 are 21, 31, and 43 nm, respectively (Fig. 1b). Their spherical shape and nearly monodisperse nature are identified by atomic force microscopy analysis (Fig. 1c) and gel permeation chromatography (Supplementary Fig. 1). This further supported the formation of single chain polymer nanomaterials without any additional modification, formulation, or chain collapse.

To investigate the in vivo fate of nanomaterials, we initially studied the pharmacokinetic behaviour of SMPNs. Tritium labelled SMPNs were injected intravenously (i.v.) in mice (Balb/c) ($n = 4$) and the concentration of SMPNs in plasma was measured (Fig. 1d). The circulation half-lives ($t_{1/2\beta}$) and pharmacokinetic parameters of SMPNs were obtained by fitting the data using a two-compartment open model (Fig. 1b, 1d and Supplementary Table 2)[22]. Given that SMPNs have similar surface chemistry, the data supported that molecular weight of the SMPNs has a dominant role on vascular residence time. SMPN-1 showed $t_{1/2\beta}$ of 65.5 ± 5.7 h, while SMPN-3 and -9 generated 57.8 ± 1.4 h and 22.1 ± 2.6 h respectively (Fig. 1b and Supplementary Table 2). As per our knowledge, the circulation half-life of SMPN-1 ($t_{1/2\beta} = 65$ h) is the highest reported to date for any single molecule nanomaterial systems, including nanogels[23], stealth liposomes[24], micellar systems[25], nanoparticles[26], dendrimers or hyperbranched polymers[20,25], and PEGylated systems in healthy mice[27]. Remarkably, SMPN-1 showed ultra-long circulation without the need of additional modifications such as the conjugation of hydrophilic polymer chains to reduce the non-specific interactions or 'self' peptides. This suggests their inherent ability to minimize opsonization and evade immune mediated clearance. The pharmacokinetic parameters, such as elimination constants ($k_2$), and area under the curve versus time plot ($AUC_{0\to\infty}$), further confirmed the ultra-long circulation of SMPN-1 compared to the other SMPNs (Fig. 1b and Supplementary Table 2).

**Biodistribution and clearance of SMPNs.** After demonstrating the long circulation of selected SMPNs, we examined their biodistribution over a 7-day period. Accumulation of SMPNs in organs was determined by measuring the residual radioactivity of digested organs (Fig. 2a). The accumulation remains low (1.89–11.85% injected dose/g of organ) in various organs; the distribution of SMPNs showed a good correlation with their $t_{1/2}$. The short circulating SMPN-9 accumulated more in the liver (****$p < 0.0001$), spleen (*$p < 0.05$), and kidney (*$p < 0.05$) compared to the ultra-long circulating SMPNs after 144 h, however, no significant differences were observed in lungs. Irrespective of their $t_{1/2}$, the accumulation of SMPNs in the kidney and lung was decreased from 1 to 144 h and this trend was opposite to that of spleen (Fig. 2a). These highly functionalizable (containing ~14, 500–89,000 hydroxyl groups/per molecule) ultra-long circulating SMPNs with minimal organ accumulation is an addition to the field of nanomaterials and could be a potential alternative to other masking or camouflaging nanomaterials.

To further understand the elimination route and tissue localization of SMPNs, we used confocal imaging of tissue slices from organs collected at various time points after intravenous (i.v.) injection of fluorophore-labeled SMPNs in Balb/c mice (Fig. 2b, c, Supplementary Figs. 3 and 4, and Supplementary Table 3). A set of images collected from tissue sections of different parts of the organ ($n = 40$) was used for the distribution analysis of SMPNs. The distribution of SMPNs in the organs showed marked differences with respect to their $t_{1/2}$. The quantitative estimation based on the fluorescence method showed some differences with the biodistribution data obtained from the radio-labelling method (Fig. 2a–c); this is possibly due to the localized accumulation of SMPNs in the organs which is reflected in the

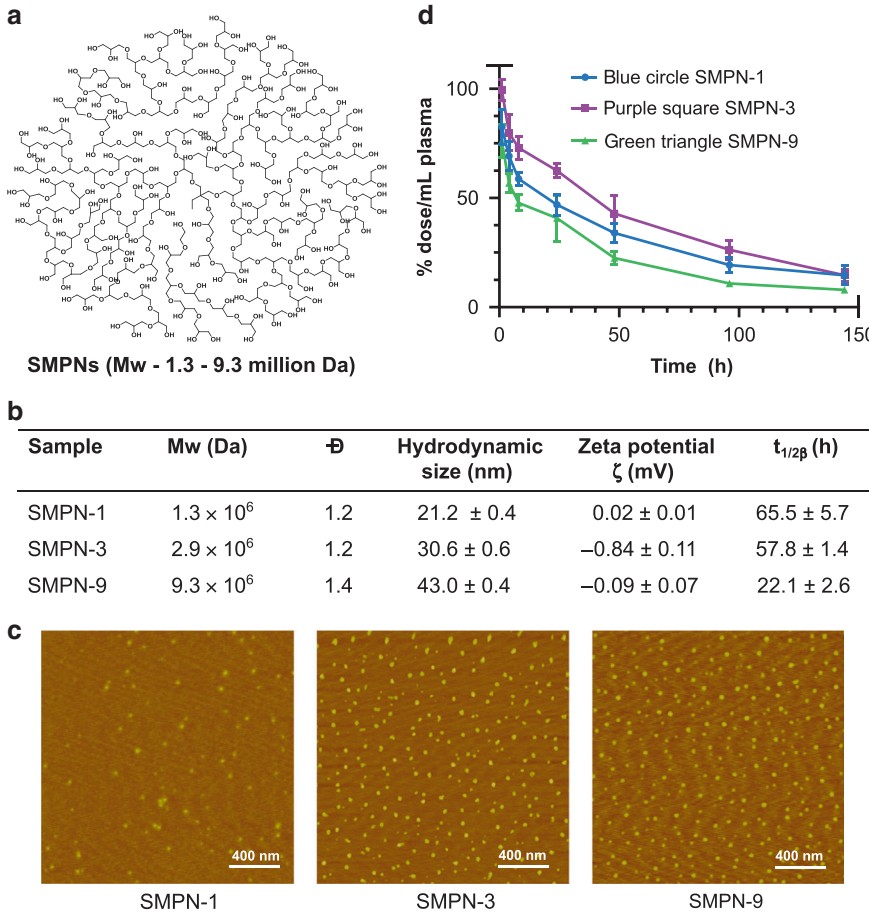

**Fig. 1 Design, characterization, and pharmacokinetics of SMPNs. a** A schematic representation of SMPNs generated by anionic ring opening polymerization of glycidol. **b** Physical characteristics, including absolute molecular weight, molecular weight distribution (Đ), hydrodynamic size, and zeta potentials in saline ($n = 3$ independent experiments) and circulation half-lives ($t_{1/2\beta}$) of SMPNs based on a two-compartment intravenous bolus model are reported. **c** Atomic force microscopy images of SMPNs (repeated two times). **d** Plasma concentration of tritium labeled SMPNs in mice ($n = 4$) after intravenous injection is shown. All the data is reported as mean ± s.d. with their respective measurements.

fluorescence quantification of tissue sections unlike complete tissue digestion in the latter case. In the liver, the ultra-long circulating SMPN-1 was distributed homogeneously throughout the organ with increased accumulation over time (*$p = 0.022$, 8 vs. 48 h), while the short circulating SMPN-9 was selectively localized, however, increased accumulation was observed over the time (*$p = 0.0078$, 8 vs. 48 h) (Fig. 2b, c and Supplementary Figs. 3 and 4). In kidney, SMPN-9 accumulated in both cortical (***$p = 0.00016$ vs control) and medullary (**$p = 0.0029$ vs control) regions within 8 h which was decreased over 48 h. In kidney, SMPN-1 is, in fact, accumulating higher than saline controls at all time points (cortex: **$p = 0.0065$ (8 h), *$p = 0.0146$ (48 h); medulla **$p = 0.0033$ (8 h), *$p = 0.0105$ (48 h) vs saline control). Although more accumulation was found at 24 h than control, there was no statistical difference found (Fig. 2b, c and Supplementary Figs. 3 and 4). In the spleen, increased amount of SMPN-1 (*$p = 0.037$ (8 h), *$p = 0.014$ (48 h) vs control) are visible in the white pulp, a region of rich in immune cells (Fig. 2b, c and Supplementary Figs. 3 and 4) with time. Contrastingly, significant SMPN-9 accumulation in the white pulp region was observed only at 8 h (*$p = 0.165$) and decreased thereafter. The marked differences in organ accumulation of these SMPNs further suggest their distinct interactions within biological systems. SMPNs were detectable in peripheral blood leukocytes up to 48 h after injection, however, there was no dependence on their uptake with respect to time and $t_{1/2}$

(Fig. 2d, e, and Supplementary Fig. 5c). This may suggest minimal involvement of circulating macrophages in the differential elimination of SMPNs from blood circulation.

**Protein corona on SMPN-1 in mice with time**. We hypothesized that the striking difference in circulation profiles for SMPNs in vivo might be originating due to the differences in the protein opsonins at the nano-biointerface of SMPNs. To investigate this, we first evaluated the composition of in vivo protein corona on SMPN-1 over different time scales using unbiased label-free quantitative mass spectrometry. The SMPN-1 was isolated from mice ($n = 4$, included both female and male mice, each biological replicate has three technical replicates) at different time points after i.v. injection. The protocol for isolation of SMPNs was initially validated with human plasma experiments (Supplementary Fig. 5a). The protein content on SMPNs was significantly different from pure human plasma that subjected to the same centrifugation/isolation protocol (Supplementary Fig. 5b). We employed the same sequential centrifugation protocol for isolation of SMPNs from mouse plasma collected from in vivo experiments. The isolation of SMPNs was further validated by fluorescence measurements after collecting fluorophore (HiLyteTM Fluor 647 amine dye) conjugated SMPNs from mice at 8 h (Supplementary Table 4 and Supplementary Fig. 6). Finally, the samples were digested in situ, and proteomics analysis was performed. Protein

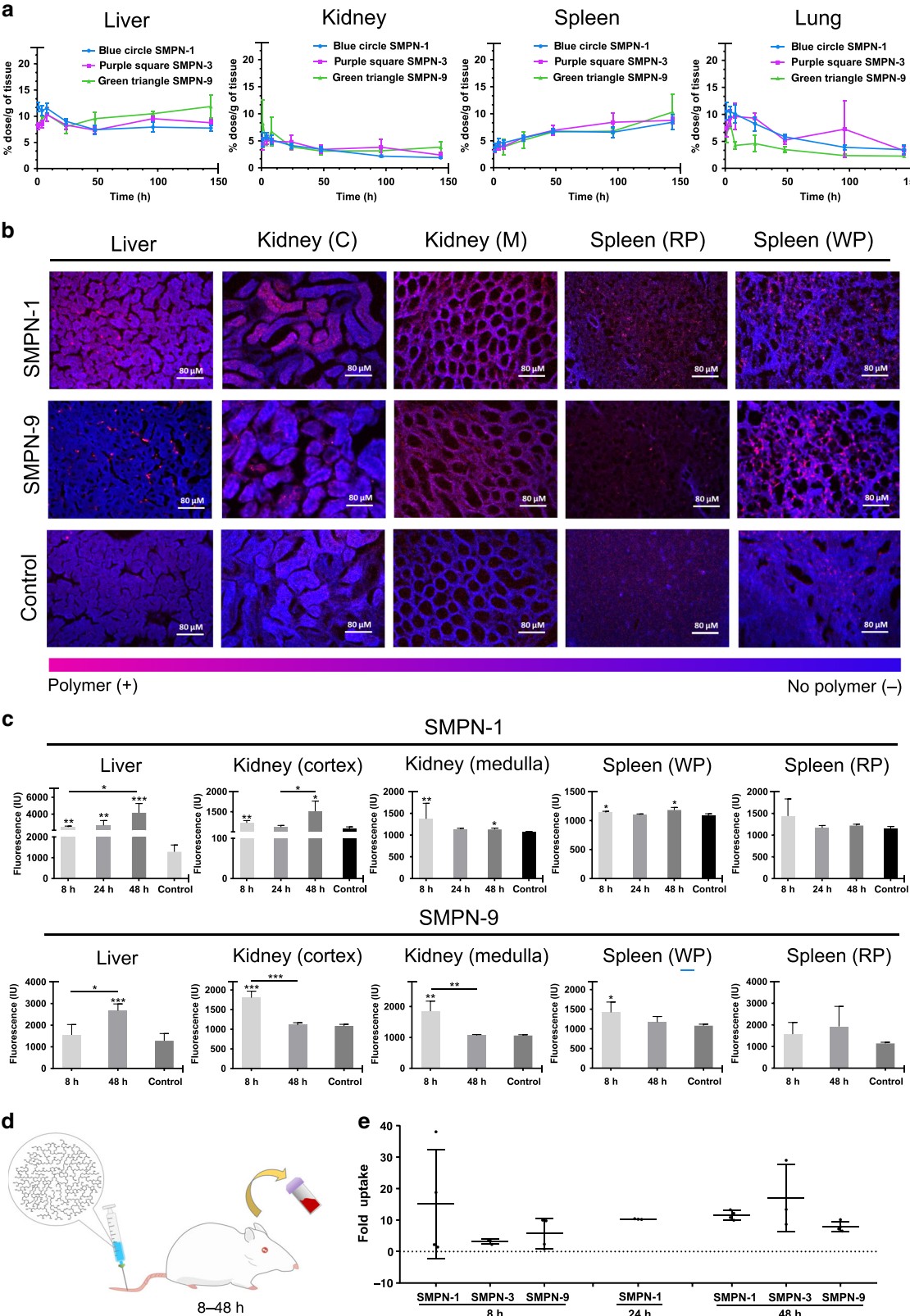

identification of protein corona on SMPNs were manually assigned through searching the mouse taxon of UnitprotKB database. Percent of abundance for each protein was calculated using label-free quantification (LFQ) intensities relative to the total sum of protein LFQ intensities for each group. Technical replicate profiles from each biological sample were averaged. We

took snapshots of protein composition on SMPN-1 at 8, 24, and 48 h, with as limited as 0.001% relative total protein content detected on its bio-interface. Within the group, correlation analysis of the biological replicates performed in Perseus software demonstrated good replicate correlation as depicted in the binary scatterplots and LFQ intensity histograms (Supplementary

**Fig. 2 Analysis of bioaccumulation of SMPNs in mice. a** Percentage of injected dose per gram of tissue (mean ± s.d.) of SMPNs over time is shown for the liver, spleen, kidney, and lungs in mice ($n = 4$) after i.v. injection. Statistical analysis was conducted by two-way ANOVA using Tukey post hoc analysis. SMPN-9 showed significantly more accumulation in the liver and spleen compared to SMPN-1 (liver: ****$p < 0.0001$; kidney: *$p < 0.05$; spleen: *$p < 0.05$). **b** Representative confocal images of tissue slices of organs at 8 h post i.v. injection of fluorophore-labelled SMPN-1 and -9 (40 images were collected for each group (10 images per mice, $n = 4$ mice). **c** Fluorescence quantification (mean ± s.d.) demonstrating the organ distribution of SMPNs from confocal microscopy images taken of 10 representative images per mouse, with $n = 4$ mice (total of 40 images). One-sided $t$-tests were performed to compare time-point values with non-injected controls. SMPN-1 was distributed homogenously in the liver across all time points, whereas in the kidney showed preferential accumulation in the cortex and medulla at 8 and 48 h (cortex: **$p = 0.0065$, *$p = 0.0146$; medulla **$p = 0.0033$, *$p = 0.0105$). In liver, SMPN-9 was selectively localized and increased with time (*$p = 0.007826$, 8 vs 48 h) and in kidney, SMPN-9 was accumulated in both cortical (***$p = 0.000157$ vs control) and medullary (**$p = 0.00219$ vs control) regions within 8 h and decreased by 48 h (cortex-***$p = 0.000219$, 8 vs 48 h; medulla ***$p = 0.002955$, 8 vs 48 h). In spleen, increased amount of SMPN-1 (*$p = 0.0372$ (8 h) and *$p = 0.0135$ (48 h) vs control) and SMPN-9 were visible in the white pulp region at 8 h (*$p = 0.046187$). **d, e** Uptake of fluorophore (HiLyteTM Fluor 647 amine) conjugated SMPNs by peripheral blood leukocytes (10,000 cells per animal, $n = 4$ mice) at 8, 24, or 48 h post i.v. injection of fluorescently labeled SMPNs in mice. The fold uptake (mean ± s.d., $n = 4$ mice) is reported and was quantified by flow cytometry analysis of isolated leukocytes. We performed a one-sided $t$-test but no significant differences were found (**e**).

Figs. 7–10). The list of top 25 most abundant proteins and the complete list of proteins on SMPN-1 from four biological replicates (three technical replicates for each biological sample) were shown (Supplementary Table 5 and Supplementary Data 1). The raw data set is available via ProteomeXchange with identifier PXD018958 [https://doi.org/10.25345/C5NX3V].

Both qualitative and quantitative changes in protein composition at the nano-biointerface of SMPN-1 were observed with time in circulation. The identified proteins on SMPN-1 were grouped according to their biological functions and were shown as relative percentages in Fig. 3a. A major portion of the protein opsonins at 8 h was composed of coagulation proteins followed by tissue leakage proteins, acute phase reactants, lipoproteins, complement, and immunoglobulins (Fig. 3a). Each time point showed changes in protein abundance and composition (Fig. 3b and Supplementary Tables 6 and 7). Over the time, the coagulation and complement proteins were decreased and immunoglobulin proteins, acute phase reactants, and lipoproteins were increased in abundance (Fig. 3c–g). Apparently, protein corona at 24 h was quite distinct from the rest of the time points across all the functional protein groups (Fig. 3c–g). A pictorial representation of changes in major group of proteins on SMPN-1 with time in circulation is detailed in Figures 3c to 3h. This analysis highlights a few important points, including the highly dynamic nature of nano-biointerface in vivo and the evidence of a continuous remodeling process at the nano-biointerface both qualitatively and quantitatively. This incessant process might be aiding the generation of long blood residency of SMPN-1.

**Comparison of protein corona on different SMPNs with time**. We next investigated the composition of protein opsonins on three SMPNs to decode if there is a correlation between blood residency and the functional role of proteins at the nano-biointerface. Protein corona snapshots of SMPNs were obtained after their isolation from mice. The list of complete set of proteins and top 25 most abundant proteins identified on SMPNs at 8 and 48 h (Supplementary Data 1 and Supplementary Tables 8 and 9). Fig. 4 summarizes the protein corona of all SMPNs (8 and 48 h) based on biological function and the analysis of common as well as unique proteins with respective to nanoparticle and time (Supplementary Tables 10 and 11 and Supplementary Data 1). We observed quantitative changes in the top-25 proteins (Supplementary Fig. 11). As shown in Fig. 4a, distinguishable changes in protein composition were observed on SMPNs with different $t_{1/2}$ even though their surface chemistry is similar. The evolution of protein corona on ultra-long circulating and short circulating SMPNs is remarkably different if we look into the details of protein fingerprints at the nano-biointerface in terms of biological

function as well as the molecular weight of the proteins (Figs. 4a and 4b). The relative percentage of proteins of common opsonins, including complement proteins, and coagulation proteins were decreased on SMPN-1 from 8 to 48 h. Over the time, the adsorption of high molecular weight proteins (>200 kDa) was increased on SMPN-3 and decreased for SMPN-1 and 9. Unlike SMPN-9, proteins having molecular weights in the range 150–200 kDa were increased on SMPN-1 and 3 with time. The proteins with 80–100 kDa showed a reverse trend with increased protein content on SMPN-9 over the time. No change was observed for proteins in the molecular weight range 50–60 kDa on SMPN-9 whereas it was decreased for SMPN-1 and 3 (Fig. 4b).

Unique proteins were identified on different SMPNs with time in circulation (Figs. 4c–4d and Supplementary Table 11). Protein snapshots at 8 h showed that there were 107 proteins common to all SMPNs, however, 52, 8, and 2, distinct proteins were identified on SMPN-1, −3, and −9 respectively (Fig. 4c and Supplementary Tables 10–11). There were more similarities between SMPN-1 and −3 than SMPN-1 and −9. SMPN-9 showed markedly different composition with only 2 and 4 common proteins between SMPN-3 and SMPN-1, respectively (Fig. 4c). A similar trend was seen at 48 h (Fig. 4d). However, the unique proteins on SMPN-1 and 3 were significantly decreased but the unique proteins moderately increased on SMPN-9. The common proteins between all the SMPNs were increased from 107 to 185, and the common proteins between SMPN-1 and 3 were decreased from 126 to 67 (Fig. 4d). These variations in unique as well as common proteins identified on different SMPNs in circulation further reinforce the fact that the protein composition at the nano-biointerface is highly dynamic in vivo and remodeling is a continuous process (Figs. 4c and 4d). These differences in protein composition might be contributing to the elimination of SMPN-9 from circulation in comparison to SMPN-1. Importantly, the observed relative protein composition at the nano-biointerface of SMPNs was not simply an expression of protein abundance in pure plasma (Supplementary Fig. 12). Apparently, these kinetics are not solely explained by mere quantitative changes as described in the literature[8] and even simple Vroman effect could not solely explain the varied compositional fingerprint over the time as well as with the molecular weight of SMPNs. We believe that soft hydrophilic nature of the interface on SMPNs generates such loosely bound protein corona and thus, facilitating the remodeling process.

To further look into the details of the proteins, we analyzed the protein corona with respect to their biological function, (Fig. 5a–f). In general, coagulation proteins constitute a major portion of the protein corona at 8 h regardless of their $t_{1/2}$ but the protein content was decreased with time (Fig. 5a). The presence of a large amount of coagulation proteins at the nano-biointerface strongly suggests that studies using anticoagulated plasma or serum may not be ideal

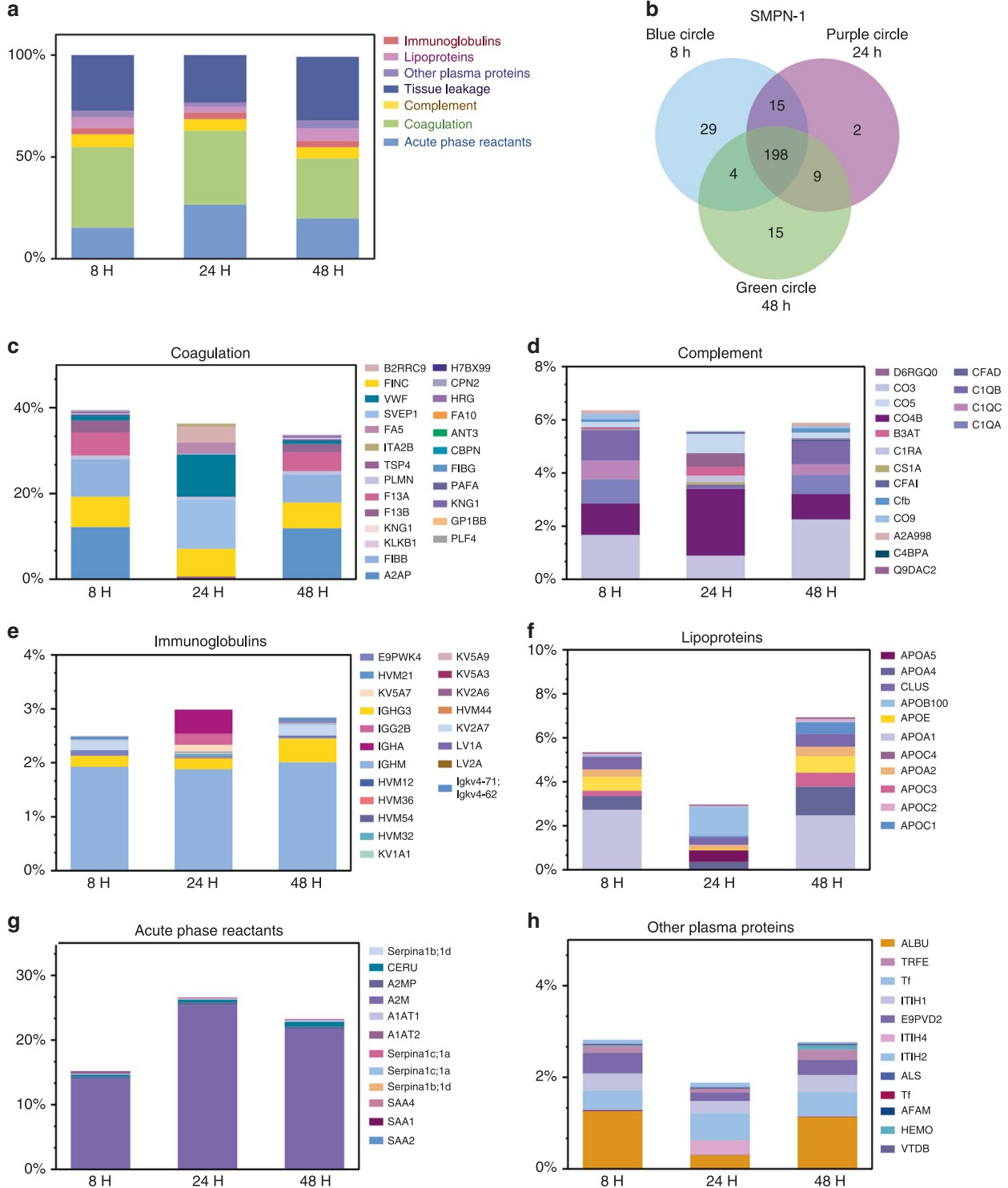

**Fig. 3 Analysis of protein corona formed at the nano-biointerface of SMPN-1.** Proteins identified on SMPN-1 at 8, 24, and 48 h post i.v. injection in mice ($n = 4$) were classified based on its biological function. **a** Abundance of each functional protein group as the percentage of total proteins on SMPN-1 shown at each time point. **b** The unique and common proteins between 8, 24, and 48 h time on SMPN-1 depicted as a Venn diagram. The number of proteins is shown here is an additive combination of four independent biological analyses with three technical replicates for each biological sample. Abundance of proteins classified based on their biological activity on the corona of SMPN-1 **c** coagulation proteins, **d** complement proteins, **e** immunoglobulins, **f** lipoproteins, **g** acute phase reactants, and **h** other plasma proteins.

for comparing nanomaterial's in vitro characteristics with its in vivo behaviour as in the former case, the coagulation system was inhibited and in the latter case, most of the coagulation proteins were removed. Complement proteins decreased for SMPN-1 and increased for SMPN-3 and 9 (Fig. 5b). Irrespective of SMPNs,

immunoglobulins and acute phase reactants were increased over 48 h (Fig. 5c and 5e). For SMPN-1 and 3, the abundance of lipoproteins was increased, and it was decreased for SMPN-9 (Fig. 5d). The plasma components showed different behaviour; they increased over the time for SMPN-3 and substantially decreased for

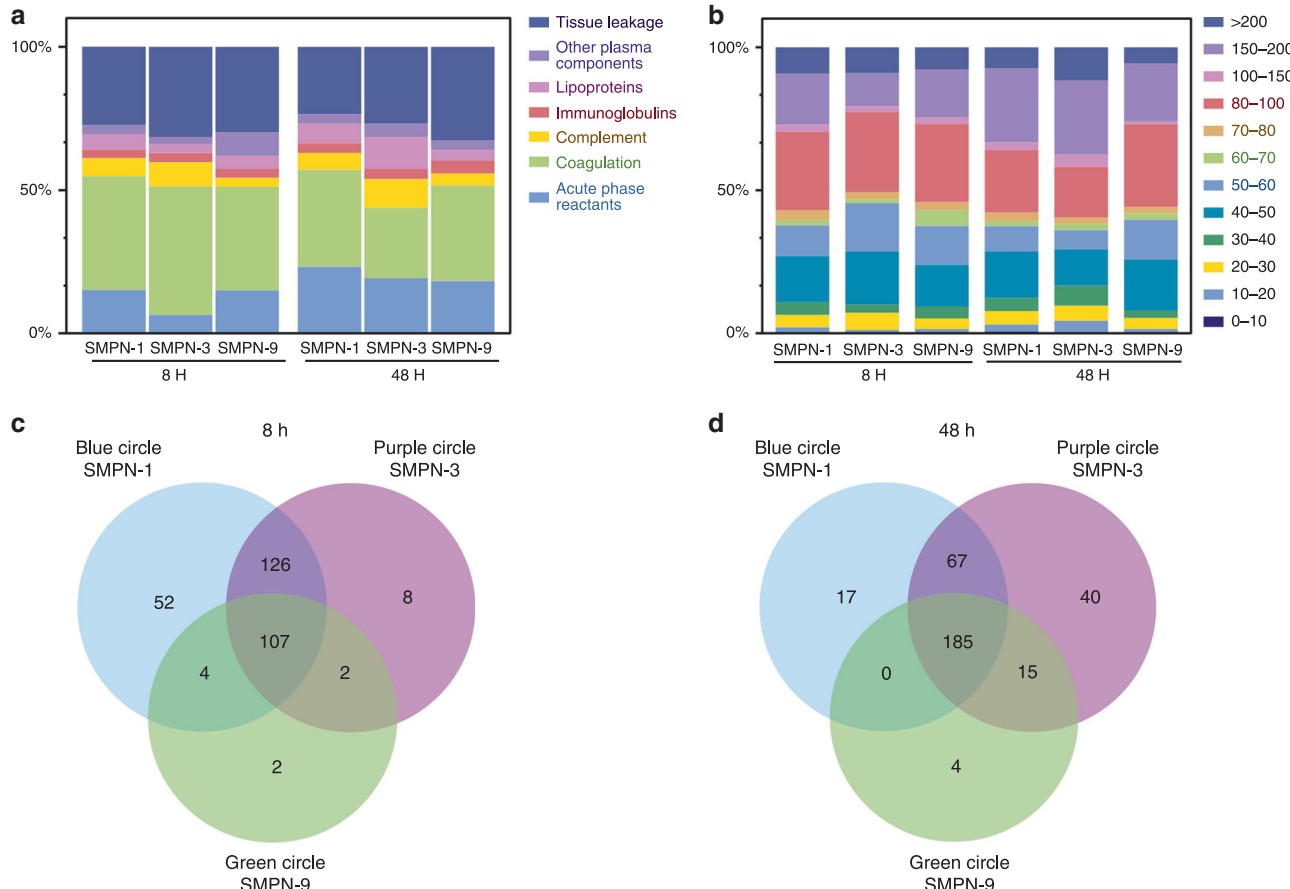

**Fig. 4 Analysis of protein corona formed at the nano-biointerface of SMPN-1, -3, and -9.** Adsorbed proteins on the surface of different SMPNs at 8 and 48 h post i.v. injection in mice (n = 4) were determined. Abundance of each functional protein group as the percentage of total proteins on SMPNs with circulation time (**a**) and molecular weight (**b**). Unique proteins identified at the nano-biointerface of SMPNs with time on different SMPNs at 8 and 48 h post i.v. injection in mice. Venn diagrams depicting unique and common proteins identified on different SMPNs at **c** 8 h and **d** 48 h post-injection in mice.

SMPN-9 (Fig. 5f). No visible difference was found for SMPN-1. Taken together, these data demonstrated the dynamic nature of protein opsonins on SMPNs in vivo circulation and the remodeling of nano-biointerface with respect to its blood residency.

Apparently, the collective protein fingerprint at the nano-biointerface is dictating the SMPN's residence in the blood compartment or its clearance, rather than the contribution or role of a specific class of adsorbed proteins[8,11,16]. Based on our current data, we believe that a dynamic protein flux may be essential for opsonization or immune recognition of soft nanomaterials in healthy mice (Fig. 6). The protein corona dynamics might depend on the unique characteristics of the system, for instance hard or soft particles, hydrophilic or hydrophobic, and charged or neutral, and it would be also influenced by species and/or phenotype of the host. Interestingly, we observed initial hints on sex differences on protein corona on SMPNs, however, further investigation is needed to validate this (Supplementary Fig. 13). Importantly, the current data may not be generalized for all types of nanomaterials. Further investigation into this dynamic protein flux at the biointerface are highly recommended to identify the protein flux on different types of nanomaterials, predict the biological outcome, and investigate the interactions in various diseases conditions. Further studies using this new model are needed to understand if any specific functional protein groups or proteins could influence the formation of dynamic protein flux. Such studies might offer a unique opportunity to better design novel materials, which evade immune recognition with minimal toxicity and have long blood residence.

In summary, we reported a class of single molecule polymer nanomaterials with varying residency in mice as a relevant model to improve our understanding on interactions at the nano-biointerface in vivo. Our analysis on protein composition of nanomaterials isolated from mice at different time points (hours to days) confirmed that the protein composition at the biointerface is highly dynamic and remodelled while in circulation. The remodelling of protein opsonin composition at the nano-biointerface may be the key for long blood residence time or faster clearance from circulation. SMPNs that release initially bound common opsonins can evade the immune system and can reside in blood for longer time periods. We believe the soft and hydrophilic nature of the current nanomaterials result in less stable protein interaction at the interface and may be contributing to the observed protein corona remodelling. The data presented here will have important implications in the field of nanotoxicology of nanomaterials and provide insights into the designing safe and immune system evading polymeric nanomaterials.

## Methods

**Materials.** All the reagents and chemicals were purchased from Sigma–Aldrich (Oakville, Ontario) and used without further purification unless otherwise mentioned. Deuterated solvents (D$_2$O and MeOD, 99.8% D) were purchased from Cambridge Isotope Laboratories, Inc. HiLyteTM Fluor 647 amine dye was purchased from Anaspec (Fremont, California). Glycidol was purified by vacuum distillation at 45 °C and stored over flame dried molecular sieves (4 Å) at 4 °C under argon. Tritiated methyl iodide solution in toluene was purchased from ARC Radiochemical (St. Louis, MO) and used it after dilution in anhydrous dimethyl-sulfoxide. NMR spectra ($^1$H, $^{13}$C, and inverse-gated (IG) $^{13}$C) were recorded on a

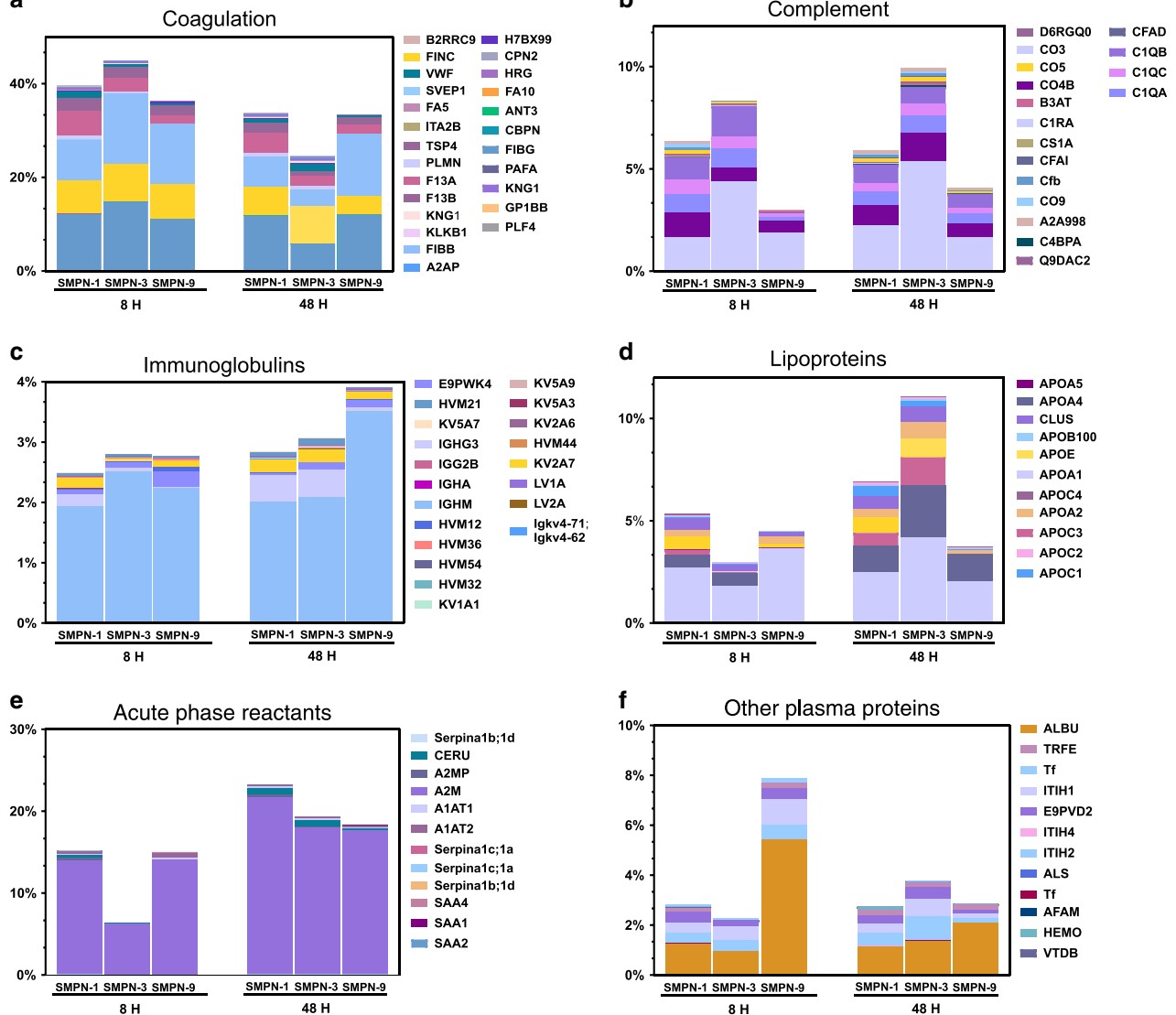

**Fig. 5 Classification of adsorbed proteins with respect to their biological function.** Abundance of protein groups as the percentage of total proteins on SMPNs classified based on their biological activity on various SMPNs at 8 and 48 h; **a** coagulation proteins, **b** complement proteins, **c** immunoglobulins, **d** lipoproteins, **e** acute phase reactants, and **f** other plasma proteins.

Bruker Avance 300 and 400 MHz NMR spectrometers. Degree of branching was measured in deuterated water ($D_2O$) with a relaxation delay of 6 s, using an equation, $DB = 2D/(2D + L)$, where D and L represent the intensities of the signals corresponding to the dendritic and linear units respectively[28]. Absolute molecular weights of the polymeric nanomaterials were determined by gel permeation chromatography (GPC) on a Waters 2695 separation module fitted with a DAWN HELEOS II multiangle laser light scattering (MALS) detector coupled with Optilab T-rEX refractive index detector, both from Wyatt Technology, Inc., Santa Barbara, CA. GPC analysis was performed using Waters ultra-hydrogel columns (guard, linear and 120) and 0.1 N $NaNO_3$ buffer (pH = 7.0) was used a mobile phase, and dn/dc value for nanomaterials used is 0.12 mL/g. Zeta potential measurements were performed on zetasizer (Malvern). Fluorescence measurements were performed on a Varian Cary Eclipse Fluorimeter (Agilent Technologies).

**Synthesis of SMPNs.** All the SMPNs were synthesized according to the following protocol[21]. The macroinitiator, hyperbranched polyglycerol (HPG), was synthesized according to the following protocol[28]. A representative synthetic protocol for SMPN-9 was given here. All the reaction steps were performed under an inert atmosphere. The macroinitiator, HPG (Mw-840 kDa, Đ-1.2) (2.5 G, 0.034 mols of total OH groups) was dissolved in anhydrous MeOH (5.0 mL) and made a thin film around the walls of flame dried three neck round bottom flask and dried the polymer under vacuum at 75 °C for 24 h to completely remove any minute amounts of water and methanol. Anhydrous DMF (35 mL) was added to the dried polymer and make sure that it was completely soluble. To this solution, KH suspension in oil (30 %) (targeted 10% hydroxyl groups for deprotection, 80 mg, 1 eq)

was added slowly. Reaction temperature was raised to 95 °C, stirred for 30 min to ensure all the macroinitiator was completely dissolved. To this homogenous solution, dried glycidol (51 mL) was added slowly (1.4 mL/h) after connecting the flask to an overhead stirrer (stirring speed-200 rpm). After glycidol addition was completed, stirring of the reaction mixture was continued for another 6 h, then cooled to RT and quenched slowly with 0.01 M HCl. The polymer was dissolved in methanol and precipitated in acetone (precipitated two more times). The precipitate was further dissolved in water and purified by dialysis (RC dialysis membrane MWCO-50,000 Da) against water for 5 days (water replacements for every 8 h). The SMPN-9 was stored as an aqueous solution at 4 °C (yield-74%). The SMPN-9 was characterized by NMR and GPC-MALS. A similar protocol was used for the synthesis of SMPN-3 and SMPN-1 except that the amount of glycidol used (see the details for the amount of glycidol used in different experiments, Supplementary Table 1).

**Conjugation of fluorophore to SMPNs.** 1,2-Diol groups (<1%) of the SMPNs were converted into aldehydes by treating SMPNs with $NaIO_4$ (1 eq) in water using our published protocol[29]. The reaction mixture was stirred at RT for overnight and dialyzed against water for 24 h using dialysis membrane (MWCO-50, 000 Da, water replacements for every 6 h). SMPNs were labeled with HiLyteTM Fluor 647 Amine dye (Anaspec) via reductive amination and the resultant product was reduced with $NaCNBH_3$ (3 eq). Finally, the remaining free aldehydes were quenched using ethanolamine (20 eq) and dialyzed against water using a cellulose dialysis membrane for 48 h (water replacements for every 8 h, RC dialysis membrane MWCO-50,000 Da). SMPN-dye conjugates were characterized by fluorescence spectroscopy

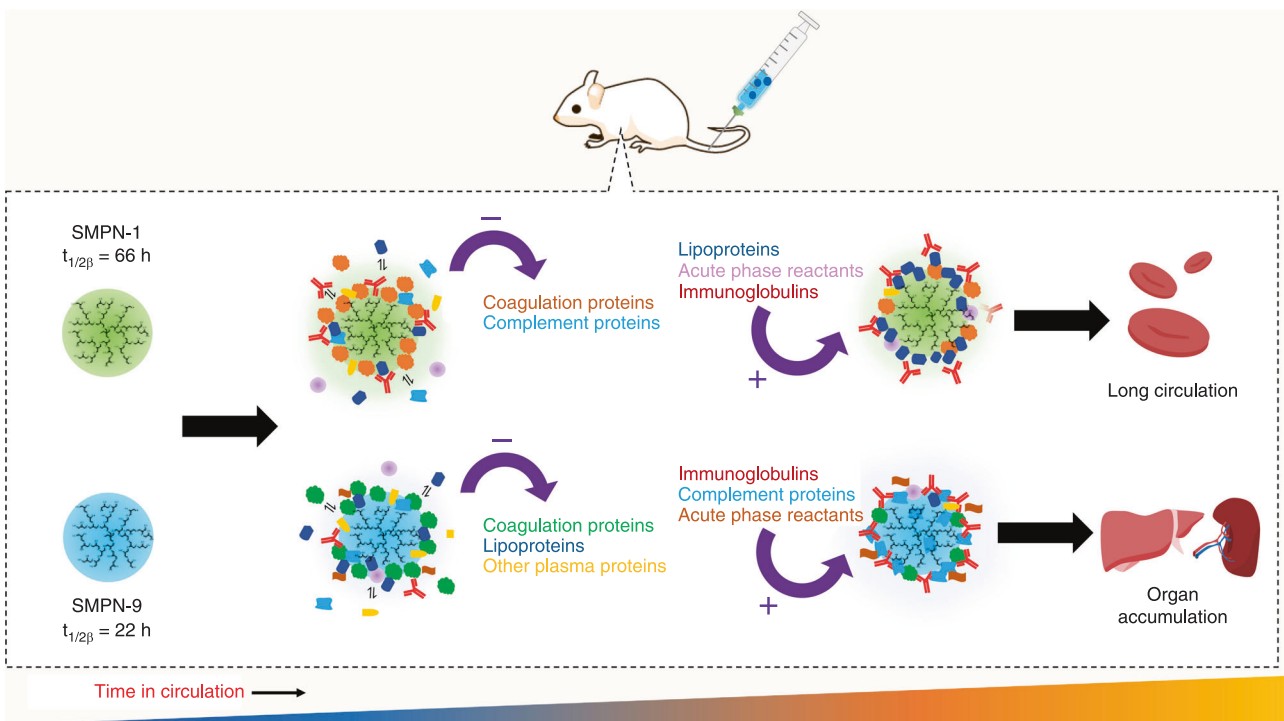

**Fig. 6 A pictorial representation of formation dynamic protein flux on SMPNs.** Coagulation and complement proteins that were absorbed initially on the surface of SMPN-1 were released during the circulation and accumulation of lipoproteins and acute phase reactants happened over 48 h on the surface. In contrast, short-circulating SMPN-9 showed with high abundance of complement proteins and acute phase reactants at 48 h.

to make sure that all the conjugates have loaded with a similar quantity of dye (Supplementary Table 3).

**Determination of hydrodynamic size and charge of SMPNs.** The hydrodynamic size of the SMPNs were measured in 0.1 M NaNO₃ buffer using multi angle laser light scattering (MALS) detector (DAWN HELEOS II) coupled with Quasi-Elastic Light Scattering (QELS) detector from Wyatt Technology, Inc., Santa Barbara, CA. Zeta potential of SMPNs was obtained using Malvern Zetasizer (Nano ZS90, He-Ne laser 633 nm) and using disposable folded capillary cells. For zeta potential measurements, three sets of measurements consisting of 100 runs each were conducted in 150 mM NaCl solution at 25 °C. Count rates were 187.4, 145.3, and 39.5 kcps, respectively, for SMPN-1, -3, and -9.

**Determination of morphology of SMPNs.** The morphology of SMPNs was determined using atomic force microscopy (AFM). SMPNs were dissolved in water at a concentration between 0.05 and 0.1 mg/mL. Ten microliters of the solution was dropped onto a cleaned Si wafer and dried overnight. The morphology of the deposited SMPNs was acquired in tapping mode in air with using a silicon probe (spring constant of 42 N/m and frequency of 320 kHz) and AFM with multimode Nanoscope IIIa controller (Digital Instruments, Santa Barbara, CA), equipped with an atomic head of $130 \times 130 \ \mu m^2$ scan range.

**Radiolabelling of SMPNs.** Radiolabelling of SMPNs (SMPN-1, 3, and 9) was performed according to the following protocol[30]. Briefly, the dried SMPN-1 (11 mg, Mw- $1.3 \times 10^6$, 100 nmol) was dissolved in dry DMSO (5 mL) under argon and NaH (0.3 mg, 140 μmol) was added. After stirring the solution at room temperature for 2 h, C³[H]₃I (100 μL) was added to methylate around 1% of the hydroxyl groups. The reaction mixture was stirred for another 20 h at room temperature and quenched the reaction mixture by the addition of water (2 mL). Tritiated SMPNs were purified by dialysis against water (RC dialysis membrane MWCO 1000) until the radioactivity of dialysate reached very minimal (50–100 dpm). The labeled nanomaterial solution was were filtered through 0.2 μm syringe filter, and concentration (mg/mL) was determined by weighing the dry nanomaterials after freeze drying the known volume (50 μL) of SMPNs. The specific activity of the SMPN was measured by scintillation counter. The osmolarity of the SMPN was adjusted by adding appropriate amount of NaCl and used it for pharmacokinetic and biodistribution studies. SMPN-3 and SMPN-9 were also labeled using similar protocols.

**Circulation half-life and biodistribution of SMPNs.** Female Balb/c mice ($n = 4$, 6–8 weeks) were injected intravenously (bolus) via lateral tail vein with a solution of tritiated SMPNs at a concentration of 1 mg/mL (four mice per group) at the prescribed dose of 20 mg/kg. Mice were sourced from Envigo. Mice were housed in

cages in normal thermoneutral temperatures (between 24 and 26 °C) under stable 50% humidity conditions using light dark cycles of 12/12. The injected volume was 200 μL per 20 g mouse. Mice were terminated at different time points (1, 4, 8, 24, 48, 96, and 144 h) by CO₂ inhalation, and blood was collected by cardiac puncture. Plasma was isolated by centrifuging the blood samples at 2000 g for 10 min. Aliquots of plasma (50 μL) were analyzed for their radioactivity by scintillation counting. Major organs, including liver, spleen, kidney, heart, and lung were removed from all the animals after the termination process, weighed, and processed for radioactivity measurements. Livers were made into a 30% homogenate in a known amount of water using a polytron tissue homogenizer. All other organs were dissolved in 500 μL Solvable®. Aliquots (in triplicates) of 200 μL of the organ solutions were transferred to scintillation vials and the vials were incubated at 50 °C overnight or up to a few days until completely dissolved, then cooled to room temperature prior to addition of 200 mM EDTA (50 μL), 10 M HCl (25 μL) and 30% H₂O₂ (200 μL). This mixture was incubated for 1 h at RT prior to addition of scintillation cocktail (5 mL). Radioactivity of the samples was measured by scintillation counting. The circulation half-lives ($t_{1/2\beta}$) and pharmacokinetic parameters of SMPNs were obtained by fitting the data using a two-compartment open model[29].

**Isolation of SMPNs from human plasma.** Huma plasma was isolated from a healthy donor from whole blood collected in citrate tubes by centrifugation for 2000 g for 15 min. Next, plasma was centrifuged at 10,000 g for 15 min in order to remove debris and microparticles. The isolation protocol is as described for our in vivo experiments where SMPNs (500 mg/kg, or approximately 6.25 mg/mL) were incubated in human plasma for 1 h at 37 °C. SMPNs were isolated from plasma fractions by ultracentrifugation (Beckman Coulter Optima Centrifuge, TLA 100.3 rotor) at 202,507 g for 1.5 h[31]. After initial centrifugation, the supernatant was removed and the pellet was washed with saline solution (0.5 mL). Centrifugation (1.5 h) and resuspension (0.5 mL, at physiological pH) were repeated five times. Finally, the SMPN samples were reconstituted in saline (0.2 mL, at physiological pH) and protein content on the samples (5 μL) was measured using Nanodrop™ Spectrophotometer (Thermo Scientific ND-2000) on protein A280 mode (Supplementary Table 5).

**SMPNS' isolation from mice after intravenous administration.** Mice (female and male, Balb/c, $N = 4$, 6–8 weeks) were administered an intravenous injection of SMPN-1, 3, or 9 at a dose of 500 mg/kg. Mice were sourced from Envigo. Mice were housed in cages in normal thermoneutral temperatures (between 24 and 26 °C) under stable 50% humidity conditions using light dark cycles of 12/12. Blood samples were collected at 8, 24, and 48 h for mice injected with SMPN-1 and at 8 and 48 h for mice injected with SMPN-3, and 9. The time points were selected based on the circulation time and to obtain a higher amount of isolated SMPNs for

analysis. After the initial isolation of plasma fractions from the blood samples as described above and subsequent centrifugation at 10,000 g for 15 min to remove debris, a sequential centrifugation and washing protocol was used to isolate SMPNs. SMPNs were isolated from plasma fractions by ultracentrifugation at 200,000 g for 1.5 h[31]. After initial centrifugation, the supernatant was removed and the pellet was washed with saline solution (0.5 mL). Centrifugation (1.5 h) and resuspension (0.2 mL, at physiological pH) was repeated five times. Finally, the SMPN samples were reconstituted in saline (0.2 mL, at physiological pH) and protein content on the samples (5 μL) was measured using NanoDrop™ UV-Vis Spectrophotometer (Thermo Scientific ND-2000) on protein A280 mode (Supplementary Table 5). Equal amounts of protein were taken from the suspension for proteomics analysis. Samples were digested for immediately for proteomic analysis.

To validate our SMPN isolation protocol, SMPNs were labeled (for protocol see conjugation of fluorophore to SMPNs) with HiLyteTM Fluor 647 amine dye and injected them in mice. After the isolation (8 h) of the nanoparticles from the plasma, fluorescent measurements of the conjugated SMPNs were performed to detect the presence of our SMPNs in the isolated suspension. Fluorescence measurements were performed for isolated SMPNs to make sure that SMPNS are isolated with proteins (Supplementary Fig. 5) and samples were digested for immediately for proteomic analysis.

**LC-MS analysis of tryptic digests**. Samples (10 μg) were incubated with dithiothreitol alkylated (0.25 μg, 30 min, 37 °C) with iodoacetamide (1.25 μg, 30 min, 37 °C), and then digested with trypsin (0.25 μg, 18.5 h in 37 °C) followed by extraction at a 1:50 protein:enzyme ratio, essentially as described[32]. Peptides (10 μG) were acidified (one volume 1% TFA), desalted on a high capacity C18 STAGE tip[33], and solubilized in 0.1% formic acid. 2 μg of digested peptides were analyzed on nanoflow-LC-MS/MS system (Bruker Impact II Q-Tof, with Proxeon EasyLC system, featuring in-house packed 400 mm × 50 μm integrated emitter columns, containing C18 stationary phase ReproSil-Pur 120 C18-AQ 3 μm (Dr Maisch, Ammerbuch-Entringen, Germany) and run with 90 min H₂O:ACN gradients. The LC C18 columns included a fritted trap column with, pulled-tip and a 50-cm analytical column produced and packed in-house. Peptides were separated using a 70 min linear gradient of increasing Buffer B. Buffers A and B were 0.1% formic acid and 0.1% formic acid and 80% acetonitrile, respectively. Data were acquired with the instrument set to scan from 200 to 2000 m/z, 100 μs transient time, 10 μs prepulse storage, 7 eV collision energy, 1500 Vpp collision RF, $a+2$ default charge state (i.e., if charge state could not be assigned, it was assumed to be $+2$), intensity-dependent MS/MS acquisition rates ranged from 4 to 16 Hz, 3.0 s cycle time, and the intensity threshold was 250 cts. Raw data was searched against the Uniprot Mouse proteome (uniprot.org) using MaxQuant (v.1.5.3.30)[34]. MaxQuant search settings are included: trypsin cleavage specificity, one allowed missed cleavage, fixed carbamidomethyl modification, variable oxidated methionine and N-terminal acetylation, 0.07 Da precursor mass tolerance, 40 ppm fragment mass tolerance, and 1% protein and peptide FDR calculation based on reverse hits. Label-free quantitation (LFQ) was enabled (with min ratio count 1) and used for intensity comparisons[35]. The mass spectrometry proteomics data have been deposited to the ProteomeXchange Consortium via the PRIDE partner MassIVE repository (UCSD, San Diego, CA, USA) with the data set identifier: PXD018958. The corresponding ProteomeXchange details are available at http://proteomecentral.proteomexchange.org/cgi/GetDataset?ID=PXD018958.

**ProteomeXchange submission details**. Project Name: Blood circulation of soft nanomaterials is governed by dynamic remodeling of protein opsonins at nanobiointerface.

Project accession code: PXD018958
Project https://doi.org/10.25345/C5NX3V.

**Protein identification and reproducibility analysis**. Protein identification of protein corona on SMPNs was manually assigned through searching the mouse taxon of UnitprotKB database. Peptide fragment hits pertaining to other species and fragments of mouse hemoglobin were removed as contaminants from the isolation protocol. Percent abundance for each protein was calculated using label-free quantification (LFQ) intensities relative to the total sum of protein LFQ intensities for each group. Average profile data from three technical replicates are shown for four different mice (biological replicates) are shown for each group. Reproducibility analysis (scatterplots and Pearson correlation between biological replicates for each treatment group at each time point) was conducted on Perseus Data Visualization Software by MaxQuant. Protein hits from false positive and known common contaminants were removed. Data was log(2) transformed. Proteins were considered quantifiable if able to be quantified at least three times with an LFQ intensity of 20 in at least 1 treatment group. Missing values were imputed into the data table from a normal distribution with an imputation width 0.3 and downshift 1.8 and plotted on multi-scatter plots and histograms.

**SMPN accumulation and uptake in organs and cells**. SMPN accumulation and uptake in organs and cells was further evaluated using fluorescence-based measurements. Female Balb/c mice ($N = 4$, 6–8 weeks) were given an intravenous injection of fluorophore-conjugated SMPN-1 and SMPN-9 at a dose of 500 mg/kg.

Mice were sourced from Envigo. Mice were housed in cages in normal thermoneutral temperatures (between 24 and 26 °C) under stable 50% humidity conditions using light dark cycles of 12/12. Blood and organs (liver, kidney, spleen and lung) were harvested at 8, 24, and 48 h and stored in 10% formalin for 4 °C for up to 1 week prior to processing for histology. 10 μm cryosections of liver, kidney and spleen were prepared for assessment of SMPN accumulation in organs by confocal microscopy (Zeiss III Spinning Disk Confocal). We have selected representative images and were included in quantification (in total, $n = 40$). All the images have been shown at the same magnification with scale bars included. Buffy coat fractions were isolated from the blood samples and washed 3 times with PBS. A Cytoflex flow cytometer (Beckman Coulter) was used to identify the leukocyte population (10,000 cells per animal analyzed) stained with FITC-labelled mouse anti-human CD45 antibody (Immunotech; cat. no. IM0782U, 1:40 dilution) and assess uptake of labelled SMPNs in vivo.

**Human blood sample collection**. Blood was collected from healthy and consenting donors at the Centre for Blood Research with protocol approval from the University of British Columbia clinical ethics committee. Whole blood was collected in 3.8% sodium citrate coated tubes (BD VacutainerTM buffered citrate sodium (0.105 M; 9:1 blood/anticoagulant)). Serum was collected in silica spray coated serum tubes (BD VacutainerTM Plus Plastic Serum) and prepared by leaving tubes undisturbed to allow for clot formation for 30 min at room temperature, followed by centrifugation at 2000 g for 15 min in an Allegra X-22R centrifuge (Beckman Coulter, Canada). Platelet rich plasma (PRP) was collected by centrifuging citrated whole blood samples at 150 g for 12 min. Platelet poor plasma (PPP) was collected by centrifuging whole blood samples at 2000 g for 15 min. Red blood cell (RBC) suspensions were prepared by washing packed red cells with PBS for four times and resuspension in PBS to yield a 20% hematocrit cell suspension. We followed our published protocols to study the blood compatibility of the SMPNs[20,36].

**Assessment of hemolysis**. Hemolysis was assessed using the Drabkin's reagent assay for the determination of hemoglobin release from lysed red cells. SMPN solutions prepared in saline were incubated with whole blood or washed RBC suspension (1:9 v/v SMPN:blood) for 1 h at 37 °C. Whole blood incubated with water (1:15 v/v blood:water) was used as a positive control. Saline-incubated whole blood or RBC suspension was used as a normal control. Samples were then added in duplicates to a 96-well plate containing Drabkin's reagent. Next, samples were centrifuged at 12,000 g for 1 min to collect and add supernatant to wells. Upon mixing with the reagent hemoglobin is rapidly converted to a cyano derivative, measured spectrophotometrically at 540 nm. Absorbance readings obtained were used to calculate percentage of hemolysis after incubation with each SMPN sample. Independent studies with three different donors were conducted.

**Assessment of blood coagulation**. Assessment of clotting time via intrinsic coagulation pathway was done through the activated partial thromboplastin time (aPTT) test using the Stago coagulation analyzer (ST4 Diagnostica Stago). SMPN samples were incubated with PPP (1:9 v/v polymer:plasma) for 1 h at 37 °C to obtain final SMPN concentrations of 0.1, 1.0 and 10.0 mg/mL. Saline-incubated PPP was used as a normal control. After incubation, innovin® partial thromboplastin reagent (Dade Behring) was added to the SMPN-PPP mixture and the solution was transferred to cuvette strips. Samples were heated to 37 °C for 2 min before the final addition of calcium chloride (CaCl₂) to initiate clotting. Measurements were done in triplicate and repeated in three different donors.

**Assessment of complement activation**. Activation of the complement system leads to the eventual formation of the terminal complement complex (TCC, or SC5b-9) that mediates the cell lysis occurring in response to an antigen. Measurement of SC5b-9 was done through enzyme immunoassay (MicroVue). SMPN samples prepared in saline were incubated with plasma (1:9 v/v) for 1 h at 37 °C. After incubation, SMPN-incubated in platelet poor plasma samples were diluted using specimen diluent reagent (1:200) and 100 μL was added to each microassay well. Specimen diluent was used as a blank. Plates were incubated with the samples in room temperature for 1 h and washed three times with prepared wash buffer. After washing, 50 μL of SC5b-9 conjugate was added to each well and incubated at room temperature for 30 min, followed by another set of washes. After incubation with the conjugate, 100 μL of the substrate was added to each well and incubated at room temperature for 15 min. In all, 100 μL of stop solution was added and the absorbance measurement at 450 nm was taken within 30 min of halting the reaction. Pre-prepared standards were used to generate a standard curve. Measurements were done in duplicate and tested in two different donors.

**Analysis of toxicity of SMPN-1 in mice**. Histological examination of tissues was performed on liver, kidney, and spleen for a SMPN-1 at 8, 24, and 48 h in experiments described in the section of SMPN accumulation and uptake in organs and cells. All organs were fixed in 10% formalin fixed organs, parafilm embedded, and sectioned. All the organ sections were stained for hematoxylin and eosin, and photomicrographs were captured on the Thermo Fisher EVOS XL core imaging system, at a ×20 magnification (0.868 μm/pixel conversion factor).

**Statistical analysis**. Statistical analysis was performed using GraphPad Prism 7 (Graphpad Software, San Diego, USA). Group comparisons to control groups were conducted using *t*-tests. Comparisons between molecular weight groups were performed using one-way ANOVA. If significance was determined, post-hoc multiple comparison analysis was conducted with Tukey test. Significance was determined with a corrected *α* value of 0.05. Unless otherwise stated, biocompatibility and uptake data were generated from the mean values of three independent experiments. All data is presented as mean ± s.d.

**Reporting summary**. Further information on research design is available in the Nature Research Reporting Summary linked to this article.

## Data availability

All generated data in this study is available in the Main Manuscript, Supplementary Information, or Supplementary Data 1. Mass spectrometry proteomic data, including raw data and search results have been deposited to the ProteomeXchange consortium via the PRIDE partner MassIVE repository (UCSD, San Diego, CA, USA) with the data set identifier: PXD018958. The corresponding ProteomeXchange details are available at http://proteomecentral.proteomexchange.org/cgi/GetDataset?ID=PXD018958.

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

## Acknowledgements

We thank the Macromolecular Hub, CBR, for the use of their research facilities and thank Drs. Marcel Bally and Nancy Dos Santos for help with animal studies at British Columbia Cancer Research Centre. We acknowledge the funding by Canadian Institutes of Health Research (CIHR), Natural Sciences and Engineering Council of Canada (NSERC) and Canada Foundation for Innovation (CFI). J.N.K. holds a Career Investigator Scholar award from the Michael Smith Foundation for Health Research (MSFHR). S.A. acknowledges a MSFHR postdoctoral fellowship. L.T. acknowledges funding from NSERC CGS-M and the NSERC CREATE NanoMat Program. J.R. was supported by funds from Genome Canada and Genome British Columbia (214PRO).

## Author contributions

Data was generated by S.A., L.T., A.P., K.Y., R.S., I.C., and J.R. Data analysis was performed by S.A., L.T., K.Y., R.S., I.C., and J.R. Paper was prepared by S.A., L.T. and J.K. with inputs from other authors. JK provided the grant support and supervision of the project.

## Competing interests

All authors declare no competing interests.

## Ethical approval

Blood from healthy consented donors was either collected at Centre for Blood Research, University of British Columbia. The protocol was approved by clinical ethical committee of the University of British Columbia. The animal studies were conducted at the Experimental Therapeutics Laboratory at the British Columbia Cancer Research Centre, Vancouver, Canada. The protocol was reviewed and approved by the Institutional Animal Care Committee (IACC) at the University of British Columbia.
