## [Peer Review File · Nature Communications]

Reviewers' Comments:

Reviewer #1:

Remarks to the Author:

In their manuscript entitled "The blood circulation of soft nanomaterials in vivo is governed by dynamic remodeling of protein opsonins at the nano-biointerface" Abbina et al. investigate protein corona formation on a small panel of three differently sized single molecule polymer nanomaterials.

The manuscript is overall well written and figures are nicely formatted.

Most of the claims of the paper are based on a "quantitative" proteomic analysis of nanoparticle bound proteins.

However, as it stands, the presented data by no means support the conclusions presented.

Reproducibility of the proteomics data is very low. I cannot see how any interpretation from only two biological replicates in combination with high quantitative variability (>5 fold differences even for proteins in the Top10 most abundant) can be valid.

Materials and Methods:

Nanoparticle isolation:

The authors must prove that isolation of SMPNs is possible with this protocol. The SMPNs will have a density very similar to proteins. It may have escaped the authors that centrifugation at 100.000xg will pellet all kinds of larger (lipo-)protein complexes from plasma.

Negative controls (without SMPNs) are missing. No buffer compositions for isolation / wash steps are provided. No volumes are provided.

In my opinion, the authors are pelleting "something large from plasma" but by no means specifically nanoparticles or their interacting proteins.

Why were the particles stored for a week – apparently without protease inhibitors - at 4°C before subsequent proteomic analysis? This will likely lead to unwanted sample degradation and negatively affect data quality.

LC-MS Analysis:

Many parameters are missing. Digestion conditions, desalting conditions, nanoLC column material etc.

Figure 1d: I cannot reproduce the half-life times reported by the authors.

Time values for 50% doses are:

SMPN-1 :24h

SMPN-3: 40h

SMPN-9: 8h

Time values for 25% doses are:

SMPN-1 :90h

SMPN-3: 120h

SMPN-9: 48h

Initial half-life times seem to be a lot shorter than the values reported by the authors, and based on Figure 1d, half-life time of SMPN-3 must be longer than that of SMPN-1

Figure 2b:

There are extreme differences in tissue structure. Scalebars are missing. All conditions for each tissue must be shown at same magnification

Figure S1

The authors claim to see a difference between SMPN-1 and SMPN-3, but I doubt that the slight shift in elution time can explain a 2.2-fold size difference?

Figure S3:

The data for kidney look most strange. How can SMPN-1 accumulate at 8h and 48h, but not 24h? For spleen, it looks like SMPN-1 accumulate at 8h, but 24h and 48h look even less intense than the negative control?

Figure S4: Negative Controls and 24h timepoints are missing.

Why does Kidney tissue structure for SMPN-9 look completely different at 8h compared to 48h? There seems to be an issue with the spleen (WP) image series – tissue structures seem completely different.

Table S3: Calculating a standard deviation from only two measurements does not make a lot of sense.

Table S4: data for SMPN-3 and SMPN-9 are missing for 24h and 48h timepoints.

Table S5: these data are misleading. Investigating the Excel-File (Supplementary dataset1, tab labeled Dataset S6-S8) reveals that of the 364 proteins (number given by the authors in Table S5), only 244 have any non-zero value. It is completely unclear, why the authors count proteins having only zero values as identified?

Furthermore, there is a reproducibility issue. Of the 244 proteins having any non-zero value, only 106 are identified in more than one condition/replicate. This indicates either a highly unreproducible proteomic workflow or a serious problem with FDR control.

Interestingly, the samples are labeled "1.1", "2.3" and "6.5" – what do these numbers refer to? It looks like molecular weights, but those would be in stark contrast with the MW reported in the manuscript.

Table S6: Again, these data indicate a non-reproducible proteomic workflow. File (Supplementary dataset2, tab labeled Dataset S9-S11) reveals that of the 364 proteins (number given by the authors in Table S5), only 316 have any non-zero value. It is completely unclear, why the authors count proteins having a zero value as identified?

Furthermore, there is a reproducibility issue. Of the 316 proteins having any non-zero value, only 116 are identified in more than one condition/replicate. This indicates either a highly unreproducible proteomic workflow or a serious problem with FDR control.

The relative amounts between replicates are not at all reproducible, e.g. α 2-Macroglobulin varies between 43.5% (48h.R2) and 7.0% (48h.R1). Carbonic Anhydrase is rank 6 (4.6%) in replicate 1, but <1% (rank 19) in replicate 2.

Analysis must be repeated, at least three biological replicates must be analyzed. Each biological sample must be analyzed in at least three technical replicates.

Reviewer #2:

Remarks to the Author:

This is a nice and informative paper demonstrating the dynamic nature of protein corona during its in vivo residency time. I would publish this paper after major revision noted below:

1- The observed variation of protein corona can be the results of analyzing the remaining nanoparticles in blood over the time and not the remodeling of protein corona. In other words, the particles with high opsonin amounts might have been removed by immune system and over the time the captured and analyzed particles were the ones that had not the opsonin participation in corona composition.

2- The number of presented proteins in Venn Diagrams of Figures 3 and 5 should have SD, if the authors performed enough repeats of the mass spect data.

3- What had been seen by the authors may be valid for the employed nanoparticles; the authors should tone down their conclusions, as their results might not be valid for other types of nanoparticles.

4- The authors are supposed to conduct the experiments on both male and female mice to consider the effect of sex as well. Recent reports revealed that both the sex have considerable effects in both biological identity and fate of nanoparticles.

5- It is now increasingly being understood that any type of disease may change the plasma composition and therefore the biological identity of nanoparticles (known as disease-specific or personalized protein corona); the author are encouraged to discuss this effect to draw the attention of the readers to the fact that reproducing the presented data may be only valid for the employed healthy mice.

Detailed response to the reviewers' comments for the manuscript "The blood circulation of soft nanomaterials *in vivo* is governed by dynamic remodeling of protein opsonins at the nano-biointerface" (Manuscript ID: NCOMMS-19-16133-T)

We thank editor and reviewers for providing their feedback, and here, we addressed all the comments raised by the reviewers.

Reviewers' comments:

Reviewer #1 (Remarks to the Author):

In their manuscript entitled "The blood circulation of soft nanomaterials *in vivo* is governed by dynamic remodeling of protein opsonins at the nano-biointerface" Abbina *et al.* investigate protein corona formation on a small panel of three differently sized single molecule polymer nanomaterials. The manuscript is overall well written and figures are nicely formatted.

Most of the claims of the paper are based on a "quantitative" proteomic analysis of nanoparticle bound proteins. However, as it stands, the presented data by no means support the conclusions presented. Reproducibility of the proteomics data is very low. I cannot see how any interpretation from only two biological replicates in combination with high quantitative variability (>5 fold differences even for proteins in the Top10 most abundant) can be valid.

Response: We thank the reviewer for the positive outlook on the manuscript presented. Your highly thoughtful and critical comments directed us to perform additional experiments and reanalyze the data to validate our conclusions. After taking consideration of comments by this reviewer on the validity and reproducibility of the proteomics data, we have now repeated our entire *in vivo* studies (N = 4 mice) and proteomics analysis. In addition, we have added the three technical replicates for each biological replicate in our proteomics analysis to demonstrate the reproducibility and confidence in the proteomics data interpretation. These extensive proteomic data are now included in the revised manuscript to substantiate our conclusions (Fig. 3-5, Supplementary Figure 5-11 and 14, and Supplementary Table 4-11). Our interpretation and conclusions remain valid.

Materials and Methods:

Nanoparticle isolation:

Comment # 1. *The authors must prove that isolation of SMPNs is possible with this protocol. The SMPNs will have a density very similar to proteins. It may have escaped the authors that centrifugation at 100.000xg will pellet all kinds of larger (lipo-)protein complexes from plasma. Negative controls (without SMPNs) are missing.*

Response: We thank the reviewer for this comment. We are very much aware about the large protein complexes present in the blood plasma, which may interfere with our analysis. Thus, we initially developed a robust protocol for the isolation of SMPNs from human plasma and then adapted this method for our *in vivo* experiments. The workflow is given in figure 1A (below).

The method we employed for our nanoparticle-protein corona isolation is well established in the literature (e.g., Docter *et al.*, *Nature Protocols*, **2014**, 9, 2030–2044, and Yu *et al.*, *ACS Nano*, **2014**, 8, 7687–7703). Similar protocols were also used for isolation of PEGylated liposomes of similar sizes and densities from mouse plasma (Giulimondi, *et al.*, *Nat. Commun.* **2019**, 10).

We adapted these methods in our current nanoparticle isolation. In our protocol, SMPNs (500 mg/kg, or approximately 6.25 mg/mL) were incubated in human plasma for one hour at 37 °C. SMPNs were isolated from plasma fractions by a sequential centrifuging protocol (Supplementary Information Section 2.2.1 and Supplementary Figure 5). Pure plasma was also subjected to the same treatment. The protein content of the isolated SMPNs was determined by NanoDrop™ UV-Vis Spectrophotometer and compared it with pure plasma (undergone similar centrifugation cycles), shown in figure 1B (below). The abundance of the protein corona on SMPNs was quite significant from pure plasma alone (**p = 0.003746, plasma Vs SMPN-3; ****p = 0.000039, plasma Vs SMPN-9).

Figure 1. The workflow of SMPN isolation. (A) SMPN isolation was performed by collecting whole blood from mice in EDTA tubes. Next, tubes were centrifuged at 2000 g for 15 minutes for plasma collection. Plasma was collected into a new tube and centrifuged at 10,000 g for 15 minutes to remove any additional microparticles and debris. Next, plasma was ultracentrifuge at 200,000 g for 90 minutes. The supernatant was removed, and the SMPN pellet was resuspended (vortexed) in saline until homogeneously distributed. This washing step was repeated five times before collecting the final pellet in 200 µL of saline. Protein collections were measured by NanoDrop™ UV-Vis Spectrophotometer, and 10 µg aliquots were collected and subjected to trypsin digest and C18 column purification before mass spectrometry analysis. (B) The SMPNs (500 mg/kg) were incubated in human plasma for 1 h and were isolated by our sequential centrifuging protocol. Pure plasma was also subjected to the same treatment. The abundance of the proteins on SMPNs was quite significant

than pure plasma alone (**p = 0.003746, plasms Vs SMPN-3; ****p = 0.000039, plasms Vs SMPN-9).

To further support the validity of the isolation protocol, we have performed the following experiment. SMPNs were fluorescently labeled with HiLyte™ Fluor 647 amine dye and injected in mice. After the isolation of the nanoparticles from the plasma, fluorescent measurements were performed to detect the presence of our SMPNs in the isolated suspension. This data is provided in the figure 2 (below) and in the supplemental information (Supplementary Figure 6). The data clearly show that fluorophore labeled SMPNs can be separated from plasma.

Figure 2. Confirmation of SMPN isolation from mouse plasma after intravenous injection. SMPN isolation was confirmed using fluorescence-based methods. Plasma collected from mice injected with Hilyte Fluor 647-labelled SMPNs were subjected to different centrifugation steps and washing protocols as described in the *Methods* section (Supporting Information Section 2.2.2). After the isolation protocol, the isolated SMPNs were suspended in saline, and fluorescent intensity was measured. The data is compared with native SMPNs (without fluorescent labelling) subjected to the same protocol and buffer control. Mean fluorescence intensity (MFI) at 670 nm and SD (N = 3 mice) are plotted. MFI from all SMPN groups (SMPN-1 ***p = 0.004; SMPN-3 **p = 0.001; SMPN-9 ***p = 0.0003) were significantly higher than the native SMPNs suggesting that SMPNs were successfully isolated by the centrifugation protocol.

In addition, we have performed the proteomics experiments of mouse plasma alone (Supplementary Figure 12), which show characteristic differences between the proteins adsorbed on SMPNs and soluble proteins present in the plasma. All these points clearly support the validity of our SMPN isolation protocol.

Comment #2. *No buffer compositions for isolation/wash steps are provided. No volumes are provided.*

Response. Thanks for bringing this point. We employed the following literature, *ACS Nano*, 2014, 88, 7687–7703, and cited it in the relevant context. However, in the revised manuscript, we have included all the details of the washing protocols and workflow figure (Supplementary Figure 5A).

The wash buffer was saline at physiological pH. After initial centrifugation, washes were conducted in 0.5 mL of the saline solution, and the final solution was resuspended in 0.2 mL saline. 15 μ L of this suspension was taken for NanoDropTM protein measurements, and equal protein amounts were taken from the suspension for proteomics analysis. We incorporated this new information in our Methods and Protocols (Supporting Information Section 2.2.2).

Comment # 3. *In my opinion, the authors are pelleting “something large from plasma” but by no means specifically nanoparticles or their interacting proteins.*

Response: With great respect, we strongly disagree with the reviewers’ point. We apologize for not providing enough data regarding the protocol used in our initial version of the manuscript. We agree that this is a crucial point to make it clear to the readership that we are isolating plasma from SMPNs rather than just mere unknown particles of plasma and reporting the data on this. We highlighted important points to rectify this and our SMPN isolation protocol is well detailed in the revised version.

Our initial SMPN isolation protocol was validated in human plasma experiments before we went ahead with mouse experiments and SMPN isolation after *in vivo* circulation. In order to ensure that our isolation protocol is indeed suitable at pelleting down the SMPNs (*ACS Nano*, **2014**, 88, 7687–7703), we further conducted additional experiments as explained in response to *comment # 1*. Our isolation protocol was further validated using fluorophore labeled SMPNs. After intravenous injection and isolation from mouse plasma, the fluorescent data of the final solution and protein content analysis for control mouse plasma without SMPNs were used to validate our protocol. The results are given in the supplementary information (Supplementary Figure 5 and 6). Please also see the response to the comment # 1.

Comment # 4. *Why were the particles stored for a week – apparently without protease inhibitors - at 4°C before subsequent proteomic analysis? This will likely lead to unwanted sample degradation and negatively affect data quality.*

Response. We thank the reviewer for raising this point. The delay was happened due to the unexpected LC-MS schedule conflicts. However, we tried our every effort to correct this issue in the revised version of the manuscript. As explained below (See response for the comment # 5), to respond to the criticism regarding the proteomics data, we have repeated our *in vivo* experiments and proteomic analysis in the revised version. We have added data from four mice in the revised version, where we performed the proteomics analysis without delay after the isolation of the SMPNs. Since we performed the analysis without storage of the isolated SMPNs in the revised version of the manuscript, we did not add protease inhibitors. Detailed information has been added in the Methods section of revised version of the manuscript.

Comment # 5. *LC-MS Analysis Many parameters are missing. Digestion conditions, desalting conditions, nanoLC column material etc.*

Response. In our initial draft, we cited an article for the method we followed. In light of the comments made by the reviewer, we have included all the details in the revised version. The following details are also provided.

Sample Preparation: Samples (10 μ g) were incubated with dithiothreitol alkylated (0.25 μ g, 30 min, 37 °C) with iodoacetamide (1.25 μ g, 30 min, 37 °C), and then digested with trypsin (0.25 μ g, 18.5 h in 37 °C) followed by extraction at a 1:50 protein:enzyme ratio, essentially as described (Foster *et al.*, Unbiased quantitative proteomics of lipid rafts reveals high specificity for signaling factors.

PNAS, **2003**, *100*, 5813–5818). 10 µg of digested sample was then cleaned up on C18 STAGE tips as described (Rappsilber *et al.*, Protocol for micro-purification, enrichment, pre-fractionation and storage of peptides for proteomics using StageTips. *Nat. Protocol*. **2007**, *2*, 1896–906).

Mass Spectrometry: 2 µg of digested peptides were analyzed on nanoflow-LC-MS/MS system (Bruker Impact II Q-Tof, with Proxeon EasyLC system, featuring in-house packed 400 mm x 50 µm integrated emitter columns, containing C18 stationary phase ReproSil-Pur 120 C18-AQ 3 µm (Dr Maisch, Ammerbuch-Entringen, Germany) and run with 90 minute H₂O:ACN gradients. The LC C18 columns included a fritted trap column with, pulled-tip and a 50-cm analytical column produced and packed in-house. Peptides were separated using a 70 min linear gradient of increasing Buffer B. Buffers A and B were 0.1% formic acid and 0.1% formic acid and 80% acetonitrile, respectively. Data were acquired with the instrument set to scan from 200 to 2000 m/z, 100 µs transient time, 10 µs prepulse storage, 7 eV collision energy, 1500 Vpp collision RF, a +2 default charge state (i.e., if charge state could not be assigned, it was assumed to be +2), intensity-dependent MS/MS acquisition rates ranged from 4 to 16 Hz, 3.0 s cycle time, and the intensity threshold was 250 cts.

Data Analysis: Data files were searched and quantified using MaxQuant software v1.6.1.0 (Cox, J. and Mann, M. MaxQuant enables high peptide identification rates, individualized p.p.b.-range mass accuracies and proteome-wide protein quantification. *Nat. Biotechnol.* **2008**, *26*, 1367–1372). The following MaxQuant search settings were included: trypsin cleavage specificity, one allowed missed cleavage, fixed carbamidomethyl modification, variable oxidated methionine and N-terminal acetylation, 0.07 Da precursor mass tolerance, 40 ppm fragment mass tolerance, and 1% protein and peptide FDR calculation based on reverse hits. Label-free quantitation (LFQ) was enabled (with min ratio count 1) and used for intensity comparisons.

Comment # 6. *Figure 1d: I cannot reproduce the half-life times reported by the authors.*

Time values for 50% doses are:

SMPN-1 :24h

SMPN-3: 40h

SMPN-9: 8h

Time values for 25% doses are:

SMPN-1 :90h

SMPN-3: 120h

SMPN-9: 48h

Initial half-life times seem to be a lot shorter than the values reported by the authors, and based on Figure 1d, half-life time of SMPN-3 must be longer than that of SMPN-1.

Response. Thanks for bringing this point to clear the ambiguity. In this manuscript, we have reported the pharmacokinetics data analyzed using a two-compartment model for extracting PK parameters. In a one-compartment model analysis, the values found by the reviewer may be considered correct, however, the one-compartment model is often not representative of drug or nanoparticle distribution in the body as it treats the body as a single homogenous volume where mixing of the compound into the compartment is instantaneous. The two-compartment model utilizes an initial distribution phase (alpha phase) and an elimination phase (beta phase) and the standard practice for determining the PK parameters of nanoparticles as described in several publications in the literature (Notari, R. E. *Biopharmaceutics and Clinical Pharmacokinetics. An Introduction*. New York, NY: Marcel Dekker

Inc.; 1980). To eliminate any ambiguity in the data interpretation, we have added additional information in the Methods section and have also included distribution phase half-lives in addition to elimination phase half-lives (Supplementary Table 2).

Comment # 7. *Figure 2b: There are extreme differences in tissue structure. Scalebars are missing. All conditions for each tissue must be shown at same magnification*

Response. We thank the reviewer for bringing this point. We have recollected the images and repeated the analysis. We have selected more representative images for figure 2b and were included in quantification (in total, N = 40, Figure 2c). All the images have been shown at the same magnification in the revised manuscript, and the scale bars were included. The conclusions remain valid.

Comment # 8. *Figure S1. The authors claim to see a difference between SMPN-1 and SMPN-3, but I doubt that the slight shift in elution time can explain a 2.2-fold size difference?*

Response. We determined the molecular weight of SMPNs using gel permeation chromatography-multiangle light scattering (GPC-MALS). We are using absolute molecular weight determination rather than referencing any standards. The method for determining the molecular weight is validated using our detector calibration (using Toluene) and the molecule weight standard human serum albumin. This method for molecular weight analysis is the most comprehensive and extensively reported in the literature (Brooks and coworkers, *Macromolecules*, 2006, 39, 7708-7717). We have provided the absolute molecular weight analysis by most comprehensive and high standard Triple Detection method gel permeation chromatography provided by Wyatt Technologies. The given traces in the original manuscripts represent the refractive index of the samples only. We provide the additional information, which includes the chromatogram from all three detectors, including light scattering (red), QELS (pink), viscometer (black), and refractive index (blue) (Figure 3, below). The peak position is not a parameter in the molecular weight determination in this analysis. In addition, conventional gel permeation chromatography cannot be used for the determination of molecular weight analysis of dendritic polymers reported in our manuscript. We are highly confident about the data provided.

Figure 3. Chromatograms are derived from triple detection gel permeation chromatography analysis (scattering (red), QELS (pink), viscometer (black), and refractive index (blue)).

Comment # 9. *Figure S3: The data for kidney look most strange. How can SMPN-1 accumulate at 8h and 48h, but not 24h? For spleen, it looks like SMPN-1 accumulate at 8h, but 24h and 48h look even less intense than the negative control?*

Response. We would like to bring the following information to reviewers' attention in response to this point. The images shown are representative images. Quantification of the analysis is reported in Fig. 2C. Our data is reported from the analysis of N = 40 images from different parts of the histological section of the organs. We are reporting here with high integrity what we observed from our analysis of several images and for drawing conclusions. SMPN-1 is, in fact, accumulating in the kidney higher than saline controls at all time points (cortex: **p=0.0065 (8 h), *p=0.0146 (48 h); medulla **p = 0.0033 (8 h), *p = 0.0105 (48 h) vs saline control). Although more accumulation was found at 24 h than saline control, there is no statistical difference was found (Fig. 2c and Supplementary Figure 3). Quantification for all time points is shown in Fig. 2c. We have observed no statistical difference between the samples. We have updated this information in the Methods section of the revised manuscript.

In the spleen, quantification shows that SMPN-1 is accumulating in the white pulp region compared to negative controls; however, no differences have been observed in the red pulp regions. It supports the potential role of immune cells in the clearance of SMPNs. However, further investigation is needed to validate it. Quantification plots in Fig. 2b have been edited to show significance compared to saline in the revised version of the manuscript.

Comment # 10. *Figure S4: Negative Controls and 24 h time points are missing. Why does Kidney tissue structure for SMPN-9 look completely different at 8h compared to 48h? There seems to be an issue with the spleen (WP) image series – tissue structures seem completely different.*

Response. We have added the following information to increase the clarity of presentation. The negative controls were initially removed for the sake of redundancy (as they were previously shown in Supplementary Figure 3 in the previous version of the manuscript). However, we have now included that in the revised Supplementary Figure 4. We thoroughly analyzed the images one more time by selecting more representative images in addition to our earlier images and increased the number of figures for quantification (N = 40). New histological images were provided where the replacements are needed, for instance, histological images for kidney (SMPN-9) at 48 hours. More representative/consistent spleen (WP) images were also taken and supplemented in the revised version.

Comment # 11. *Table S3: Calculating a standard deviation from only two measurements does not make a lot of sense.*

Response. We apologize for the confusion. This data is a combination of six data points (measured twice, and each measurement has three replicates). We re-organized it and reported as the standard deviation of six data points. It is the primary data of the fluorescence measurements of SMPNs after their conjugation with a fluorophore to make sure that all SMPNs have same amount of fluorescence w.r.t. molar concentrations.

Comment # 12. *Table S4: data for SMPN-3 and SMPN-9 are missing for 24h and 48h timepoints.*

Response. The reported data in Supplementary Table 4 is corresponding to the protein content on the isolated SMPNs from mice. We collected the SMPN-1 at 8, 24, and 48 h and SMPN-3 & 9 at 8 and 48 h only, and the protein content was measured by NanoDrop UV-Vis Spectrophotometer. We included the detailed information in the figure legend and in the table of the revised manuscript (Supplementary Table 4).

Comment # 13. Table S5: these data are misleading. Investigating the Excel-File (Supplementary dataset1, tab labeled Dataset S6-S8) reveals that of the 364 proteins (number given by the authors in Table S5), only 244 have any non-zero value. It is completely unclear, why the authors count proteins having only zero values as identified? Furthermore, there is a reproducibility issue. Of the 244 proteins having any non-zero value, only 106 are identified in more than one condition/replicate. This indicates either a highly unreproducible proteomic workflow or a serious problem with FDR control.

Response. We removed this table to clear the ambiguity. We re-formatted all the tables as per our new data. For additional information on the data analysis, see the following response for comment 15.

Comment # 14. *Interestingly, the samples are labeled “1.1”, “2.3” and “6.5” – what do these numbers refer to? It looks like molecular weights, but those would be in stark contrast with the MW reported in the manuscript.*

Response. We apologize for this typo. We corrected it and relabelled as SMPN-1, 3, and 9, respectively. Originally the sample labels were given in relation to the number average molecular weight, and thus generated the mistake.

Comment # 15. Table S6: Again, these data indicate a non-reproducible proteomic workflow. File Supplementary dataset 2, tab labeled Dataset S9-S11) reveals that of the 364 proteins (number given by the authors in Table S5), only 316 have any non-zero value. It is completely unclear, why the authors count proteins having a zero value as identified? Furthermore, there is a reproducibility issue. Of the 316 proteins having any non-zero value, only 116 are identified in more than one condition/replicate. This indicates either a highly unreproducible proteomic workflow or a serious problem with FDR control. The relative amounts between replicates are not at all reproducible, e.g. α 2-Macroglobulin varies between 43.5% (48h.R2) and 7.0% (48h.R1). Carbonic Anhydrase is rank 6 (4.6%) in replicate 1, but <1% (rank 19) in replicate 2. Analysis must be repeated, at least three biological replicates must be analyzed. Each biological sample must be analyzed in at least three technical replicates.

Response. We agree and thankful to the reviewer for providing such a critical point. To address the reproducibility issues, we have repeated the comprehensive proteomic analysis of all SMPNs for N = 4 biological replicates, and each biological replicate has three technical replicates. The data in the revised manuscript is collected from the new *in vivo* analysis and proteomic analysis. It increases the confidence in our data and validity of our conclusions. As per our knowledge, no report was found on protein corona studies for such long-circulating times. Our data is in par with any of the proteomics data published in reputed journals. Based on this new data, we have updated in the main text (Fig. 3, 4, and 5) and supporting information (Supplementary Figure 7-11 and 14 and Supplementary Table 4-11). We did not use the proteomics data given in the previous version of the manuscript as the preparation of the samples, proteomic analysis time, MS conditions, analysis, *etc.*, were changed.

To demonstrate the reproducibility of our data analysis, we have developed scatterplots and histograms for both biological replicates and found very good correlations (Supplementary Figure 7-10). We here provide the scatterplots for SMPN-1 at three time points. The Pearson correlation coefficients (~0.7-0.9) between biological replicates, validated a very high level of reproducibility of our data. The current data is one of the best data set available for the analysis of protein corona data given in the literature considering the number of analyzed proteins. To the best of our knowledge, this is the first data set with such a high correlation between biological replicates in the case of *in vivo* protein corona studies.

Figure 4. Binary scatterplots for SMPN-1 at various time points. Scatterplots demonstrating each of the mean (N = 3) LFQ intensities for each mouse (M#) compared against mice in the same group. Pearson correlation coefficients are reported in the upper left-hand corner.

Finally, we thank the reviewer for carefully reading and highlighting the critical points which allowed us to improve the quality of the data and the presentation. We strongly believe that the comments were adequately addressed.

Reviewer #2 (Remarks to the Author):

This is a nice and informative paper demonstrating the dynamic nature of protein corona during its *in vivo* residency time. I would publish this paper after major revision noted below:

Response: We thank the reviewer for his/her highly positive support for our manuscript. Your insightful comments increased the vigor in the analysis and validation of our conclusions from the data given. We repeated our studies and performed extensive analysis to support our conclusions. We also included the sex differences in the samples. We modified our conclusions based on the new data and the conclusions made in the previous version of the manuscript remain valid. We also highlighted the importance of the dynamic protein corona on nanoparticles in healthy mice.

Comment #1. *The observed variation of protein corona can be the results of analyzing the remaining nanoparticles in blood over the time and not the remodeling of protein corona. In other words, the particles with high opsonin amounts might have been removed by immune system and over the time the captured and analyzed particles were the ones that had not the opsonin participation in corona composition.*

Response. It is a great point and agree with the reviewer. However, we have the following points regarding the selection of the time points for this analysis. Ideally, one needs to have to continuous monitoring of the nanoparticle corona *in vivo* and current techniques, and technologies will not allow for detailed measurements as we reported in this manuscript. We also highlight the fact that even though we are analyzing the proteins remained on the surface, the analyses at multiple time points provide the information on the corona changes with time. We have added this information in the discussion section of the manuscript. We have chosen our initial time point (8 h) based on the following facts: 1. The long circulation half-life times ($t_{1/2\beta}$ -22 h to 65 h) of SMPNs allowed us to collect the samples with sufficient concentrations for detailed analysis. 2. We were interested in tracking the protein corona throughout the vascular residence time of nanoparticles rather than initial time points. As demonstrated previously by other researchers (*Nat. Nanotechnol.* 2013, **8**, 772–781) initial time points are important, however, there is limited data available on long circulating nanoparticles. 3. Further, we speculated that the protein corona of nanoparticles at earlier time points most likely will be soft corona, and we are particularly interested in observing any notable changes in the hard corona. Therefore, we designed our experiments to collect the SMPNs at different time points. As explained even though some nanoparticles are taken out of circulation, the analysis at different time points will provide important information on whether any changes happening on the corona of the SMPNs. Our studies clearly show that remodeling of protein corona is happening during the circulation and may be responsible for its elimination or retention in circulation.

Comment # 2. *The number of presented proteins in Venn Diagrams of Figures 3 and 5 should have SD, if the authors performed enough repeats of the mass spec data.*

Response. We apologize for the confusion in our previous version of the manuscript. In this figure, we were qualitatively, analyzing the data by combining the different proteins identified from the all mice rather than averaging them. We have updated the figure caption with more details to convey it properly to the readers. Our intention was to investigate unique and common proteins in a more qualitative manner rather than the quantification data. We presume that the unique protein corona could provide directions for future studies. Notably, the unique protein corona would be useful to understand the complex biological interactions of nanoparticles with cellular components or tissues.

Comment # 3. *What had been seen by the authors may be valid for the employed nanoparticles; the authors should tone down their conclusions, as their results might not be valid for other types of nanoparticles.*

Response. We appreciate the reviewer to bring up this valuable point. We restrained our conclusions to our systems. We initially planned for running experiments with other nanoparticle systems, including polymeric nanoparticles (polystyrene, PLGA) and metal nanoparticles such as gold and silver, to validate this novel dynamic protein corona phenomenon. However, we were handicapped with very short circulation times of these nanoparticles. In addition to this, stability and surface properties of the materials will have a significant influence on protein corona formation. Considering these factors, we are currently working to generate stable and long circulating polymeric nanoparticles based on SMPNs with different surface properties. We believe that protein corona analysis of these systems will provide more insights into this novel phenomenon with respect to other surface properties.

Comment # 4. *The authors are supposed to conduct the experiments on both male and female mice to consider the effect of sex as well. Recent reports revealed that both the sex have considerable effects in both biological identity and fate of nanoparticles.*

Response. We thank the reviewer for important point. In light of this comment, we performed additional *in vivo* studies and proteomics analysis. Currently, we have four biological replicates, two female and two male mice, and used three technical replicates for each biological sample. The current reported data is highly reproducible across the biological replicates (supported by Pearson correlation coefficients ~0.7-0.9) (Supplementary Figure 7-10)). We also noticed some differences (Supplementary Figure 14), concerning the sex differences. However, based on the current data, the difference is not significant enough to perform additional analysis on sex differences. Perhaps, it would be significant with the increase in the number of animals used. Since the focus of the current manuscript is not performing such analysis, we will focus this as a future project. We would like to investigate these differences thoroughly with detailed biodistribution and circulation time on sex differences in our future studies before making meaningful conclusions. It will be another huge study. We highly appreciate the reviewer to bring up this valuable point.

Figure 5. Parts of a whole graph depicting average percent abundance of corona proteins classified by functional group in (N = 2) female mice (A) and (N = 2) male mice (B).

Comment # 5. *It is now increasingly being understood that any type of disease may change the plasma composition and therefore the biological identity of nanoparticles (known as disease-specific or personalized protein corona); the author are encouraged to discuss this effect to draw the attention of the readers to the fact that reproducing the presented data may be only valid for the employed healthy mice.*

Response. We are entirely in agreement with the reviewer’s point. It is known that circulation half-life of nanoparticles, polymers, and liposomes are varied with the type of animal model used. As we have perfected our techniques using SMPNs and have more understanding of the current system, we are working towards expanding this work in disease models. We will revisit a discussion regarding this in a future manuscript.

Summary of revision performed:

We performed a major revision based on the reviewers’ feedback. Here are the summarized key elements of the revised manuscript,

1. Biodistribution data of the SMPNs were reanalyzed (Fig. 2b-2c and Supplementary Figure 3-4); more representative images were included in our analysis (N = 40).
2. The isolation of nanoparticles from plasma was confirmed by fluorescence measurements of nanoparticles conjugated with a fluorophore, after collecting them from mice (Supplementary Figure 5 and 6). The isolation protocols of SMPNs were also revisited with more details (Supporting information 2.2.1 and 2.2.2).
3. We increased the sample size from N = 2 to N = 4 (biological samples) for our protein corona analysis by including both male and female mice. We have also added three technical replicates for each biological samples. The nanoparticles were isolated from mice plasma soon after their collection from mice and submitted to proteomic analysis without any delay. According to our new data, we modified the Fig. 3, 4, and 5, Supplementary Figure 5-11 and 14, and Supplementary Tables 4-11. The conclusions and interpretations remain valid.

4. Pearson correlation analysis was performed across biological replicates and depicted a very high reproducibility across the biological samples (0.7-0.9, except two outliers, Supplementary Figure 7-10).

5. Our new data further validates the critical conclusions of our original manuscript, “*continuous remodeling of protein opsonins at nano-biointerface of SMPNs is a key component in controlling their vascular residency in mice.*” We further noticed the effect of sex differences on protein corona formation (Supplementary Figure 13). However, more analysis is needed to make meaningful conclusions. We believe that more vigorous investigation is needed to adapt our hypothesis to different nanoparticles, disease conditions, and the host.

Reviewers' Comments:

Reviewer #2:

Remarks to the Author:

The authors have thoroughly and precisely addressed the concerns of this reviewer. I would publish this interesting paper.

REVIEWERS' COMMENTS

Reviewer #1

1. Tables 5-11 of the supplementary information need to have the protein abundance numbers.

Response: We thank the reviewer for the comment. In this revision, we included the protein abundance values in the Tables (Supplementary Table 5, 8, and 9) unless where protein abundances were taken as additive data from 4 different mice, particularly, for instance, in the case of unique and common proteins (Supplementary Table 6, 7, 10, and 11). However, the abundance of all individual identified proteins (N = 4 mice) are given in Supplementary Data 1.

2. The authors should submit their proteomics' raw data and search results to the ProteomeXchange repository, which allow other researchers to reproduce the analyses.

Response: We thank the reviewer for the comment. We uploaded all the proteomic data in the ProteomeXchange repository (details are given below) and also, provided in the manuscript in the sections of "LC-MS analysis of tryptic digests" as well as "Data Availability". "Mass spectrometry proteomic data, including raw data and search results have been deposited to the ProteomeXchange consortium via the PRIDE partner MassIVE repository (UCSD, San Diego, CA, USA) with the data set identifier: PXD018958[<https://doi.org/doi:10.25345/C5NX3V>]. The corresponding Proteome Xchange details are available at <http://proteomecentral.proteomexchange.org/cgi/GetDataset?ID=PXD018958>."

Reviewer #2 The authors have thoroughly and precisely addressed the concerns of this reviewer. I would publish this interesting paper.